# Escaping Saddle Points with Compressed SGD

**Dmitrii Avdiukhin**
Department of Computer Science
Indiana University
Bloomington, IN 47405
davdyukh@iu.edu

**Grigory Yaroslavtsev**
Department of Computer Science
George Mason University
Fairfax, VA 22030
grigory@grigory.us

## Abstract

Stochastic gradient descent (SGD) is a prevalent optimization technique for large-scale distributed machine learning. While SGD computation can be efficiently divided between multiple machines, communication typically becomes a bottleneck in the distributed setting. Gradient compression methods can be used to alleviate this problem, and a recent line of work shows that SGD augmented with gradient compression converges to an $\varepsilon$-first-order stationary point. In this paper we extend these results to convergence to an $\varepsilon$-*second*-order stationary point ($\varepsilon$-SOSP), which is to the best of our knowledge the first result of this type. In addition, we show that, when the stochastic gradient is not Lipschitz, compressed SGD with RAN-DOMK compressor converges to an $\varepsilon$-SOSP with the same number of iterations as uncompressed SGD [25], while improving the total communication by a factor of $\tilde{\Theta}(\sqrt{d}\varepsilon^{-3/4})$, where $d$ is the dimension of the optimization problem. We present additional results for the cases when the compressor is arbitrary and when the stochastic gradient is Lipschitz.

## 1 Introduction

Stochastic Gradient Descent (SGD) and its variants are the main workhorses of modern machine learning. Distributed implementations of SGD on a cluster of machines with a central server and a large number of workers are frequently used in practice due to the massive size of the data. In distributed SGD each machine holds a copy of the model and the computation proceeds in rounds. In every round, each worker finds a stochastic gradient based on its batch of examples, the server averages these stochastic gradients to obtain the gradient of the entire batch, makes an SGD step, and broadcasts the updated model parameters to the workers. With a large number of workers, computation parallelizes efficiently while communication becomes the main bottleneck [12, 38], since each worker needs to send its gradients to the server and receive the updated model parameters. Common solutions for this problem include: local SGD and its variants, when each machine performs multiple local steps before communication [36]; decentralized architectures which allow pairwise communication between the workers [30] and gradient compression, when a compressed version of the gradient is communicated instead of the full gradient [6, 37, 27]. In this work, we consider the latter approach, which we refer to as *compressed SGD*.

Most machine learning models can be described by a $d$-dimensional vector of parameters $\mathbf{x}$ and the model quality can be estimated as a function $f(\mathbf{x})$. Hence optimization of the model parameters can be cast a minimization problem $\min_{\mathbf{x}} f(\mathbf{x})$, where $f \colon \mathbb{R}^d \to \mathbb{R}$ is a continuous function, which can be optimized using continuous optimization techniques, such as SGD. Fast convergence of compressed SGD to a first-order stationary point (FOSP, $\|\nabla f(\mathbf{x})\| < \varepsilon$) was shown recently for various gradient compression schemes [6, 37, 27, 23, 2]. However, even an exact FOSP can be either a local minimum, a saddle point or a local maximum. While local minima often correspond to good solutions in machine learning applications [21, 39, 7], saddle points and local maxima are always suboptimal and

35th Conference on Neural Information Processing Systems (NeurIPS 2021).

it is important for an optimization algorithm to avoid converging to them. In particular, [13] show that for neural networks many local minima are almost optimal, but the corresponding loss functions have a combinatorial explosion in the number of saddle points. Furthermore, [15] show that saddle points can significantly slow down SGD convergence and hence it is important to be able to escape from them efficiently.

Since finding a local minimum is NP-hard in general [5], a common relaxation of this requirement is to find an approximate second-order stationary point (SOSP), i.e. a point with a small gradient norm ($\|\nabla f(\mathbf{x})\| < \varepsilon$) and the smallest (negative) eigenvalue being small in absolute value ($\lambda_{\min}(\nabla^2 f(\mathbf{x})) > -\varepsilon_H$). When $f$ has $\rho$-Lipschitz Hessian (i.e. $\|\nabla^2 f(x) - \nabla^2 f(y)\| \le \rho\|x - y\|$ for all $x, y$), a standard choice of $\varepsilon_H$ is $\sqrt{\rho\varepsilon}$ [32], and such approximate SOSP is commonly referred as an $\varepsilon$-SOSP. While second-order optimization methods allow one to escape saddle points, such methods are typically substantially more expensive computationally. A line of work originating with the breakthrough of [20] shows that first-order methods can escape saddle points when perturbations are added at certain iterations. In particular, a follow-up [25] show that SGD converges to an $\varepsilon$-SOSP in an almost optimal number of iterations.

In this paper, we show that even *compressed* SGD can efficiently converge to an $\varepsilon$-SOSP. To the best of our knowledge, this is the first result showing convergence of compressed methods to a second-order stationary point.

## 1.1  Related Work

**Escaping from saddle points**    While it is known that gradient descent with random initialization converges to a local minimum almost surely [28], existence of saddle points may result in exponential number of steps with non-negligible probability [17]. Classical approaches for escaping from saddle points assume access to second-order information [32, 14]. Although these algorithms find a second-order stationary point in $O(\varepsilon^{-3/2})$ iterations, each iteration requires computation of the full Hessian matrix, which can be prohibitive for high-dimensional problems in practice. Some approaches relax this requirement, and instead of full Hessian matrix they only require access to a Hessian-vector product oracle [9, 1]. While in certain settings, including training of neural networks, it's possible to compute Hessian-vector products (HVP) efficiently [33, 35], such an oracle might not be available in general. Furthermore, in practice HVP-based approaches are significantly more complex compared to SGD (especially if the workers aren't communicating in every iteration in the distributed setting) and require additional hyperparameter tuning. Moreover, HVP is typically used for approximate an eigenvector computation, which may increase the number of iterations by a logarithmic factor.

Limitations of second-order methods motivate a long line of recent research on escaping from saddle points using first-order algorithms, starting from [20]. [24] show that perturbed gradient descent finds $\varepsilon$-SOSP in $\tilde{O}(\varepsilon^{-2})$ iterations. Later, this is improved by a series of accelerated algorithms [10, 1, 11, 26] which achieves $\tilde{O}\left(\varepsilon^{-7/4}\right)$ iteration complexity. There are also a number of algorithms designed for finite sum setting where $f(x) = \sum_{i=1}^{n} f_i(x)$ [34, 4, 18], or in case when only stochastic gradients are available [40, 25], including variance reduction techniques [3, 18]. The sharpest rates in these settings have been obtained by [18], [42] and [19].

**Compressed SGD**    While gradient compression may require a complex communication protocol, from theoretical perspective this process is often treated as a black-box function: a (possibly randomized) function $\mathcal{C}$ is called a $\mu$-compressor if $\mathbb{E}\left[\|\mathbf{x} - \mathcal{C}(\mathbf{x})\|^2\right] < (1 - \mu)\|\mathbf{x}\|^2$. In a simplified form, the update step in compressed SGD can be expressed as $\mathbf{x}_{t+1} \leftarrow \mathbf{x}_t - \eta\mathcal{C}(\nabla f(\mathbf{x}))$[1]. Notable examples of compressors include the following: SIGN [6] sends the sign of each coordinate, QUANTIZATION [2] splits all possible values into buckets and communicates bucket indices, TOPK preserves only $k$ largest (by the absolute value) coordinates while setting other coordinates to 0, and RANDOMK preserves $k$ random coordinates. Among them, it's not clear how to implement SIGN and TOPK in distributed settings; [23] suggest an efficient sketch-based approximate implementation of TOPK. Both RANDOMK and sketch-based TOPK are $k/d$-compressors requiring $\tilde{O}(k)$ communication.

---

[1]The actual update equation is more complicated, see Algorithm 1.

While it is known that compressed SGD converges (e.g. [27] and the works above), the convergence was shown only to a FOSP. The crucial idea to facilitate convergence is to use error-feedback [37]: the difference between the true gradient and its compression is propagated to the next iteration.

## 1.2 Our Contributions

Our main contribution is the analysis showing that perturbed compressed SGD with error-feedback can escape from saddle points efficiently. Moreover, we show faster convergence rate for a certain type of compressors and show that such compressors exist. Inspired by the ideas from [25] and [37], we present an algorithm (Algorithm 1) which uses perturbed compressed gradients with error-feedback and converges to an $\varepsilon$-second-order stationary point (see Theorem 3.1). Our main results shows that compressed SGD with RANDOMK compressor achieves substantial communication improvement:[2]

**Theorem 1.1 (Informal, Theorem 3.2 and Corollary 3.3)** *Assume that $f$ has Lipschitz gradient and Lipschitz Hessian. Let $\alpha = 1$ when the stochastic gradient is Lipschitz and $\alpha = d$ otherwise. Then SGD with RANDOMK compressor (which selects $k$ random coordinates) with $k = \frac{d\varepsilon^{3/4}}{\sqrt{\alpha}}$ converges to an $\varepsilon$-SOSP after $\tilde{O}\left(\frac{\alpha}{\varepsilon^4}\right)$ iterations, with $\tilde{O}\left(\frac{d\sqrt{\alpha}}{\varepsilon^{3+1/4}}\right)$ total communication per worker.*

Compared with the uncompressed case, the total communication improves by $\varepsilon^{-1/4}$ when the stochastic gradient is Lipschitz and by $\sqrt{d}\varepsilon^{-3/4}$ otherwise (the sharpest results for SGD are by [19] and [25] respectively). In Theorem 1.1, we heavily rely on the following property of RANDOMK: when its randomness (i.e. sampled $k$ coordinates) is fixed, the compressor becomes a linear function. For other compressors, this property doesn't necessarily hold; in this case, we show convergence with a slower convergence rate:

**Theorem 1.2 (Informal, Theorem 3.1 and Corollary 3.4)** *Assume that $f$ has Lipschitz gradient and Lipschitz Hessian. Let $\alpha = 1$ when the stochastic gradient is Lipschitz and $\alpha = d$ otherwise. Let $\mathcal{C}$ be a $k/d$-compressor requiring $\tilde{O}(k)$ communication. Then SGD with compressor $\mathcal{C}$ with $k = \frac{d\sqrt{d}\varepsilon^{3/4}}{\sqrt{\alpha}}$ converges to an $\varepsilon$-SOSP after $\tilde{O}\left(\frac{\alpha}{\varepsilon^4}\right)$ iterations, with $\tilde{O}\left(\frac{d\sqrt{d}\sqrt{\alpha}}{\varepsilon^{3+1/4}}\right)$ total communication per worker.*

Compared with the uncompressed case, the total communication improves by $\frac{\varepsilon^{-1/4}}{\sqrt{d}}$ when the stochastic gradient is Lipschitz (note that this is the only setting where the convergence improvement is conditional, requiring $\varepsilon = o(d^{-2})$) and by $\varepsilon^{-3/4}$ otherwise. Table 1 in Section 3.1 shows communication improvements for various choices of compression parameters. We outline our main techniques and technical contributions in Section 3.2 and give the complete proof in Appendix A and B.

## 2 Preliminaries

**Function Properties** For a twice differentiable nonconvex function $f \colon \mathbb{R}^d \to \mathbb{R}$, we consider the unconstrained minimization problem $\min_{\mathbf{x} \in \mathbb{R}^d} f(\mathbf{x})$. We use the following standard [25, 19, 41, 3, 43] assumptions about the objective function $f$:

**Assumption A** *$f$ is $f_{\max}$-bounded, $L$-smooth and has $\rho$-Lipschitz Hessian, i.e. for all $x, y$:*

$$|f(\mathbf{x}) - f(\mathbf{y})| \le f_{\max}, \quad \|\nabla f(\mathbf{x}) - \nabla f(\mathbf{y})\| \le L\|\mathbf{x} - \mathbf{y}\|, \quad \|\nabla^2 f(\mathbf{x}) - \nabla^2 f(\mathbf{y})\| \le \rho\|\mathbf{x} - \mathbf{y}\|$$

**Assumption B** *Access to an unbiased stochastic gradient oracle $\nabla F(\mathbf{x}, \theta)$, whose randomness is controlled by a parameter $\theta \sim \mathcal{D}$[3], with bounded variance, i.e. for all $\mathbf{x}$:*

$$\mathbb{E}_{\theta \sim \mathcal{D}}\left[\nabla F(\mathbf{x}, \theta)\right] = \nabla f(\mathbf{x}), \quad \mathbb{E}_{\theta \sim \mathcal{D}}\left[\|\nabla F(\mathbf{x}, \theta) - \nabla f(\mathbf{x})\|^2\right] \le \sigma^2$$

As shown by the above works, smoothness allows one to achieve fast convergence for nonconvex optimization problems (namely, to use the folklore descent lemma). Similarly, Lipschitz Hessian

---

[2]$\tilde{O}$ hides polylogarithmic dependence dependence on $d$ and $\varepsilon$ and polynomial dependence on other parameters.

[3]E.g. $\theta$ is a minibatch selected at the current iteration

allows one to show fast second-order convergence, since, within a certain radius, the function stays close to its quadratic approximation (see e.g. [8, Section 9.5.3]). As common in the literature, in our convergence rates we treat $L$, $\rho$ and $\sigma^2$ as constants. We also consider an additional optional assumption (see [25] for a justification):

**Assumption C (Lipschitz stochastic gradient, optional)** *For any* $\mathbf{x}, \mathbf{y}, \theta$:

$$\|\nabla F(\mathbf{x}, \theta) - \nabla F(\mathbf{y}, \theta)\| \leq \tilde{\ell}\|\mathbf{x} - \mathbf{y}\|.$$

**Gradient Compression**  Our goal is to optimize $f$ in a distributed setting [16, 29]: given $\mathcal{W}$ workers, for each worker $i$ we have a corresponding data distribution $\mathcal{D}_i$. Then the each worker has a corresponding function $f_i(\mathbf{x}) = \mathbb{E}_{\theta_i \sim \mathcal{D}_i}[F(\mathbf{x}, \theta_i)]$ and $f = \sum_{i=1}^{\mathcal{W}} f_i$. In a typical distributed SGD setting, each worker computes a stochastic gradient $\nabla F_i(\mathbf{x}, \theta_i)$ and sends it to the coordinator machine. The coordinator machine computes the average of these gradients $\mathbf{v} = \frac{1}{\mathcal{W}} \sum_{i=1}^{\mathcal{W}} \nabla F_i(\mathbf{x}, \theta_i)$ and broadcasts it to the workers, which update the local parameters $\mathbf{x} \leftarrow \mathbf{x} - \eta\mathbf{v}$ ($\eta$ is the step size).

With this approach, with increase of the number of machines, the computation can be perfectly parallelized. However, with each machine required to send its gradient, communication becomes the main bottleneck [12, 38]. There exist various solutions to this problem (see Section 1), including gradient compression, when each machine sends an approximation of its gradient. Then coordinator averages these approximations and broadcasts the average to all machines (possibly compressing it again, see discussion on TopK and Sign in Section 1.1).

Depending on the compression method, this protocol results in different gradient approximation and different communication per machine. There is a natural trade-off between approximation and communication, and it's not clear whether having smaller per-iteration communication results in smaller total communication required for convergence. The approximation quality can be formalized using the following definition:

**Definition 2.1 ([37])** *Function* $\mathcal{C}(\mathbf{x}, \tilde{\theta})$, *whose randomness is controlled by a parameter* $\tilde{\theta} \sim \tilde{\mathcal{D}}^4$, *is a* $\mu$-*compressor if* $\mathbb{E}_{\tilde{\theta} \sim \tilde{\mathcal{D}}}\left[\|\mathbf{x} - \mathcal{C}(\mathbf{x}, \tilde{\theta})\|^2\right] < (1 - \mu)\|\mathbf{x}\|^2$.

Section 1.1 provides examples of important compressors. In our analysis, we consider general and linear compressors separately, and in the latter case, we show an improved convergence rate.

**Definition 2.2** $\mathcal{C}$ *is a* linear compressor *if* $\mathcal{C}(\cdot, \tilde{\theta})$ *is a linear function for any* $\tilde{\theta}$.

One example of a linear compressor is RandomK, which sets all but $k$ coordinates of a vector to 0; it's a $k/d$-compressor [37] and can be computed easily in the distributed setting.

**Stationary Points**  The optimization problem of finding a global minimum or even a local minimum is NP-hard for nonconvex objectives [31, 5]. Instead, as is standard in the literature, we show convergence to an approximate FOSP or an approximate SOSP, see Section 1.

**Definition 2.3** *If* $f$ *is differentiable then* $\mathbf{x}$ *is an* $\varepsilon$-*First-Order Stationary Point if* $\|\nabla f(\mathbf{x})\| \leq \varepsilon$.

An $\varepsilon$-FOSP can be a local maximum, a local minimum or a saddle point. While local minima typically correspond to good solutions, saddle points and local maxima are inherently suboptimal. Assuming non-degeneracy, saddle points and local maxima have escaping directions, corresponding to Hessian's negative eigenvectors. Following [32] we refer to points with no escape directions (up to a second-order approximation) as approximate second-order stationary points:

**Definition 2.4 ([32])** *If* $f$ *is a twice differentiable* $\rho$-*Hessian Lipschitz function then* $\mathbf{x}$ *is an* $\varepsilon$-*Second-Order Stationary Point if* $\|\nabla f(\mathbf{x})\| \leq \varepsilon$ *and* $\lambda_{\min}(\nabla^2 f(\mathbf{x})) \geq -\sqrt{\rho\varepsilon}$, *where* $\lambda_{\min}$ *is the smallest eigenvalue*[5].

---

[4]E.g. for RandomK, $\tilde{\theta}$ is the set of indices of preserved coordinates.

[5]While one can consider two threshold parameters – $\varepsilon_g$ for $\nabla f$ and $\varepsilon_H$ for $\nabla^2 f$ – we follow convention of [32] which selects $\varepsilon_H = -\sqrt{\rho\varepsilon}$, which, intuitively, balances first-order and second-order variability.

**Algorithm 1:** Compressed SGD

---

**parameters:** $\eta$ – step size, $T$ – number of iterations, $r^2$ – variance of the artificial noise, $\mathcal{I}$ – the number of iterations required for escaping, $\mathcal{R}$ – escaping radius

**input** : objective $f$, compressor function $\mathcal{C}$, starting point $\mathbf{x}_0$

**output :** $\varepsilon$-SOSP of $f$

1   $\mathbf{e}_0 \leftarrow 0^d$
2   $t' \leftarrow 0$
3   **for** $t = 0 \ldots T - 1$ **do**
4     // Reset the error after $\mathcal{I}$ iterations or in case we moved far from the initial point
5     **if** $(t - t' > \mathcal{I}$ *or* $\|\mathbf{x}_{t'} - (\mathbf{x}_t - \eta\mathbf{e}_t)\| > \mathcal{R}$ **then**
6       **if** $f(\mathbf{x}_t) < f(\mathbf{x}_{t'})$ **then** $\mathbf{x}_t \leftarrow \mathbf{x}_t - \eta\mathbf{e}_t$ **else** $\mathbf{x}_t \leftarrow \mathbf{x}_{t'}$ ;
7       $t' \leftarrow t, \quad \mathbf{e}_t \leftarrow 0^d$
8     **end**
9     Sample $\xi_t \sim \mathcal{N}_d(0^d, r^2), \quad \theta_t \sim \mathcal{D}, \quad \tilde{\theta}_t \sim \tilde{\mathcal{D}}$
10    $\mathbf{g}_t \leftarrow \mathcal{C}(\mathbf{e}_t + \nabla F(\mathbf{x}_t, \theta_t) + \xi_t, \tilde{\theta}_t)$           // Compressed gradient
11    $\mathbf{x}_{t+1} \leftarrow \mathbf{x}_t - \eta\mathbf{g}_t$           // Compressed gradient descent step
12    $\mathbf{e}_{t+1} \leftarrow \mathbf{e}_t + \nabla F(\mathbf{x}_t, \theta_t) + \xi_t - \mathbf{g}_t$           // Update the error
13 **end**
14 **return** $\mathbf{x}_T$

---

An important property of points which are not $\varepsilon$-SOSP is that they are unstable: adding a small perturbation allows gradient descent to escape them [20] (similar results were shown for e.g. stochastic [25] and accelerated [26] gradient descent). In this work we show that this property holds even for SGD with gradient compression.

## 3 Algorithm and Analysis

**Algorithm**    We present our algorithm in Algorithm 1, a compressed stochastic gradient descent approach based on [37, Algorithm 1]. In order to achieve convergence to a SOSP, similarly to [25], we add artificial random noise $\xi_t$ to gradient at every iteration, which allows compressed gradient descent to escape saddle points. At every iteration $t$, we compute the stochastic gradient $\nabla F(\mathbf{x}_t, \theta_t)$. Then we add artificial noise $\xi_t$, compress the resulting value (Line 10) and update the current iterate $\mathbf{x}_t$ using the compressed value (Line 11). However, the information is not lost during compression: the difference between the computed value and the compressed value (Line 12), $\mathbf{e}_{t+1}$, is added to the gradient in the next iteration. [27] show that carrying over the error term improves convergence of compressed SGD to a FOSP. Algorithm 1 sets $\mathbf{e}_t$ to 0 (Line 7) when conditions in Line 5 hold: either we moved far from the point where the condition was triggered last time (intuitively, the condition indicates that we successfully escaped from a saddle point), or we spent a certain number of iterations since that event (to ensure that the accumulated compression error is sufficiently bounded).

**Distributed Setting**    Algorithm 1 provides a general framework for compressed SGD in distributed settings, with implementation details depending on the choice of the compressor function $\mathcal{C}$. $\theta_t$, $\tilde{\theta}_t$ and $\xi_t$ can be efficiently shared between machines using shared randomness. Each machine $i$ maintains its own local $\mathbf{e}_t^{(i)}$ which can be computed as $\mathbf{e}_{t+1}^{(i)} \leftarrow \mathbf{e}_t^{(i)} + \nabla F_i(\mathbf{x}_t, \theta_t) + \xi_t - \mathbf{g}_t^{(i)}$. Then $\mathbf{e}_t = \frac{1}{\mathcal{W}} \sum_{i=1}^{\mathcal{W}} \mathbf{e}_t^{(i)}$. Finally, the norm in Line 7 of Algorithm 1 can be efficiently computed within multiplicative approximation using linear sketches.

### 3.1 Convergence to an $\varepsilon$-SOSP

In the following statements, $\tilde{O}$ hides polynomial dependence on $L, \rho, f_{\max}, \sigma, \tilde{\ell}$ and polylogarithmic dependence on all parameters. The next two theorems present our main result, namely that compressed SGD converges to an $\varepsilon$-SOSP[6]. The first theorem addresses the general compressor case.

---

[6]see proof sketch in Section 3.2 and the full proof in Appendix B

**Theorem 3.1 (Convergence to $\varepsilon$-SOSP for general compressor)** *Let $f$ satisfy Assumptions A and B, let $\mathcal{C}$ be a $\mu$-compressor. Let $\alpha = 1$ when Assumption C holds and $\alpha = d$ otherwise. Then for Algorithm 1 with $\eta = \tilde{O}\left(\min\left(\frac{\varepsilon^2}{\alpha}, \frac{\mu\varepsilon}{\sqrt{1-\mu}}, \frac{\mu^2\sqrt{\varepsilon}}{(1-\mu)d}\right)\right)$, after $T = \tilde{O}\left(\frac{1}{\varepsilon^2\eta}\right) = \tilde{O}\left(\frac{\alpha}{\varepsilon^4} + \frac{\sqrt{1-\mu}}{\mu\varepsilon^3} + \frac{d(1-\mu)}{\mu^2\varepsilon^2\sqrt{\varepsilon}}\right)$ iterations, at least half points $\mathbf{x}_0, \dots, \mathbf{x}_T$ are $\varepsilon$-SOSP w.h.p.*

In general, convergence to an $\varepsilon$-SOSP is noticeably slower than convergence to an $\varepsilon$-FOSP: in Appendix A we show that an $\varepsilon$-FOSP can be found in $\tilde{O}\left(\frac{1}{\varepsilon^4} + \frac{\sqrt{1-\mu}}{\mu\varepsilon^3}\right)$ iterations (the result is similar to [37], but slightly more general and under weaker assumptions). The reason for such behavior is that, in the analysis of convergence to a SOSP, compression introduces error similar to that of the stochastic noise. When the stochastic gradient is not Lipschitz (i.e. Assumption C doesn't hold), the number of iterations increases by a factor of $d$ due to stochastic noise. Unfortunately, in general the compression is not Lipschitz even for deterministic gradients: consider a TOPK compression of a vector where each coordinate is 1 with small perturbations. However, if the compressor is linear (Definition 2.2), we show improved convergence rate: the last term in the number of iterations decreases by the factor of $d$.

**Theorem 3.2 (Convergence to $\varepsilon$-SOSP for linear compressor)** *Let $f$ satisfy Assumptions A and B, let $\mathcal{C}$ be a linear compressor. Let $\alpha = 1$ when Assumption C holds and $\alpha = d$ otherwise. Then for Algorithm 1 with $\eta = \tilde{O}\left(\min\left(\frac{\varepsilon^2}{\alpha}, \frac{\mu\varepsilon}{\sqrt{1-\mu}}, \frac{\mu^2\sqrt{\varepsilon}}{1-\mu}\right)\right)$, after $T = \tilde{O}\left(\frac{1}{\varepsilon^2\eta}\right) = \tilde{O}\left(\frac{\alpha}{\varepsilon^4} + \frac{\sqrt{1-\mu}}{\mu\varepsilon^3} + \frac{1-\mu}{\mu^2\varepsilon^2\sqrt{\varepsilon}}\right)$ iterations, at least half of points $\mathbf{x}_0, \dots, \mathbf{x}_T$ are $\varepsilon$-SOSP w.h.p.*

Since RANDOMK is a linear compressor, by balancing the terms we have:

**Corollary 3.3** *For RANDOMK compressor with $k = \frac{d\varepsilon^{3/4}}{\sqrt{\alpha}}$, the total number of iterations of Algorithm 1 is $\tilde{O}(\frac{\alpha}{\varepsilon^4})$ and the total communication per worker is $\tilde{O}\left(\frac{d\sqrt{\alpha}}{\varepsilon^{3+1/4}}\right)$. When Assumption C holds, the total communication for RANDOMK decreases by the factor of $\tilde{\Theta}(\varepsilon^{-1/4})$ compared with the unconstrained case [19]. Otherwise, the total communication decreases by the factor of $\tilde{\Theta}(\sqrt{d}\varepsilon^{-3/4})$ compared with [25].*

**Compressed SGD in Distributed Settings** Below we consider different scenarios to illustrate how convergence depends on the properties of the compressor. Recall that sketch-based TOPK is a $k/d$-compressor which requires $\tilde{O}(k)$ communication. Selecting $\mu = k/d$, with $k \ll d$, by Theorem 3.1 we have $\eta = \tilde{O}\left(\min\left(\frac{\varepsilon^2}{\alpha}, \frac{k\varepsilon}{d}, \frac{k^2\sqrt{\varepsilon}}{d^3}\right)\right)$. Therefore, the total number of iterations is $\tilde{O}\left(\frac{1}{\varepsilon^4} + \frac{d}{k\varepsilon^3} + \frac{d^3}{k^2\varepsilon^2\sqrt{\varepsilon}}\right)$ and the total communication is $\tilde{O}\left(\frac{k}{\varepsilon^4} + \frac{d}{\varepsilon^3} + \frac{d^3}{k\varepsilon^2\sqrt{\varepsilon}}\right)$.

Note that the above reasoning considers a worst-case scenario. However, in practice it's often possible to achieve good compression at a low communication cost due to the fact that gradient coordinates have heavy-hitters, which are easy to recover using TOPK. We formulate this beyond worst-case scenario as the following optional assumption:

**Assumption D (Optional)** *There exists a constant $c < 1$ such that for all $t$, $\mathcal{C}(\nabla F(\mathbf{x}_t, \theta_t) + \xi_t + \mathbf{e}_t)$ provides a $c$-compression and requires $\tilde{O}(1)$ bits of communication per worker.*

In other words, Assumption D means that for all computed values, $\mathcal{C}$ provides a constant compression and requires a polylogarithmic amount of communication. This assumption can be satisfied under various conditions. For example, some methods may take advantage of the situation when gradients between adjacent iterations are close [22]. In cases when certain coordinates are much more prominent in the gradient compared to others, TOPK compressor will show good performance.

**Corollary 3.4** *Algorithm 1 converges to $\varepsilon$-SOSP in a number of settings, as shown in Table 1.*

### 3.2 Proof Sketch

In this section, we outline the main techniques used to prove Theorems 3.1 and 3.2. A recent breakthrough line of work focused on convergence of first-order methods to $\varepsilon$-SOSP [20, 9, 24, 40, 25]

Table 1: Convergence to $\varepsilon$-SOSP with uncompressed SGD, with sketch-based TOPK compressor, with RANDOMK compressor, and with a constant-compressor requiring constant communication (Assumption D, beyond worst-case assumption). We considered two settings depending on whether the stochastic gradient is Lipschitz (i.e. Assumption C holds). For each setting we select the optimal $\mu$ based on our bounds. The results show that communication of SGD with RANDOMK compression outperforms that of the uncompressed SGD by $\tilde{O}(\varepsilon^{-1/4})$ when Assumption C holds and by $\tilde{O}(\sqrt{d}\varepsilon^{-3/4})$ otherwise. Based on our results, since a constant-memory constant-communication compressor is not necessarily linear, depending on $d$ and $\varepsilon$, it may converge slower than RANDOMK.

| | Setting | $\mu$ | Iterations | Total comm. per worker | Total comm. improvement |
|---|---|---|---|---|---|
| **Lipschitz $\nabla F$** | Uncompressed [19] | $1$ | $\tilde{O}\left(\frac{1}{\varepsilon^{3.5}}\right)$ | $\tilde{O}\left(\frac{d}{\varepsilon^{3.5}}\right)$ | $-$ |
| | Sketch-based TOPK | $\sqrt{d}\varepsilon^{3/4}$ $(\varepsilon = o(d^{-2}))$ | $\tilde{O}\left(\frac{1}{\varepsilon^{4}}\right)$ | $\tilde{O}\left(\frac{d\sqrt{d}}{\varepsilon^{3+1/4}}\right)$ | $\tilde{\Theta}\left(\frac{\varepsilon^{-1/4}}{\sqrt{d}}\right)$ |
| | RANDOMK | $\varepsilon^{3/4}$ | $\tilde{O}\left(\frac{1}{\varepsilon^{4}}\right)$ | $\tilde{O}\left(\frac{d}{\varepsilon^{3+1/4}}\right)$ | $\tilde{\Theta}\left(\varepsilon^{-1/4}\right)$ |
| | Constant-memory $c$-compressor | $c > 0$ | $\tilde{O}\left(\frac{1}{\varepsilon^{4}} + \frac{d}{\varepsilon^{2}\sqrt{\varepsilon}}\right)$ | $\tilde{O}\left(\frac{1}{\varepsilon^{4}} + \frac{d}{\varepsilon^{2}\sqrt{\varepsilon}}\right)$ | $\tilde{\Theta}\left(\min(d, \frac{1}{\sqrt{\varepsilon}})\right)$ |
| **non-Lipschitz $\nabla F$** | Uncompressed [25] | $1$ | $\tilde{O}\left(\frac{d}{\varepsilon^{4}}\right)$ | $\tilde{O}\left(\frac{d^{2}}{\varepsilon^{4}}\right)$ | $-$ |
| | Sketch-based TOPK | $\varepsilon^{3/4}$ | $\tilde{O}\left(\frac{d}{\varepsilon^{4}}\right)$ | $\tilde{O}\left(\frac{d^{2}}{\varepsilon^{3+1/4}}\right)$ | $\tilde{\Theta}\left(\varepsilon^{-3/4}\right)$ |
| | RANDOMK | $\frac{\varepsilon^{3/4}}{\sqrt{d}}$ | $\tilde{O}\left(\frac{d}{\varepsilon^{4}}\right)$ | $\tilde{O}\left(\frac{d\sqrt{d}}{\varepsilon^{3+1/4}}\right)$ | $\tilde{\Theta}\left(\sqrt{d}\varepsilon^{-3/4}\right)$ |
| | Constant-memory $c$-compressor | $c > 0$ | $\tilde{O}\left(\frac{d}{\varepsilon^{4}}\right)$ | $\tilde{O}\left(\frac{d}{\varepsilon^{4}}\right)$ | $\tilde{\Theta}\left(d\right)$ |

has developed a comprehensive set of analytic techniques. We start by outlining [25], which is the sharpest known SGD analysis in the case when the stochastic gradient is not Lipschitz.

Let $\mathbf{x}_0$ be an iterate such that $\lambda_{\min}(\nabla^2 f(\mathbf{x}_0)) < -\sqrt{\rho\varepsilon}$, and $\mathbf{v}_1$ be the eigenvector corresponding to $\lambda_{\min}$. Consider sequences $\{\mathbf{x}_t\}$ and $\{\mathbf{x}'_t\}$ starting with $\mathbf{x}_0$ which are referred to as *coupling sequences*: their distributions match the distribution of compressed SGD iterates (i.e. both sequences can be produced by Algorithm 1), and they share the same randomness, with an exception that their artificial noise has the opposite sign in the direction $\mathbf{v}_1$. The main idea is that such artificial noise combined with SGD updates ensures that projection of $\mathbf{x}_t - \mathbf{x}'_t$ on $\mathbf{v}_1$ increases exponentially, and therefore at least one of the sequences moves far from $\mathbf{x}$. After that, one can use an "Improve or localize" Lemma which states that, if we move far from the original point, then the objective decreases substantially.

If we have an access to a deterministic gradient oracle and the objective function is quadratic, then gradient descent behaves similarly to the power method, since in this case: $\mathbf{x}_{t+1} = (I - \eta\nabla^2 f(\mathbf{x}_0))\mathbf{x}_t$. Adding artificial noise guarantees that there is a non-trivial projection of $\mathbf{x}_t - \mathbf{x}'_t$ on direction $\mathbf{v}_1$, and the power method further amplifies this projection. In general, the SGD behavior deviates from power method due to: 1) the difference between $f$ and its quadratic approximation and 2) stochastic noise. [25] show that the errors introduced by these deviations are dominated by the increase in direction $\mathbf{v}_1$, and therefore SGD successfully escapes saddle points.

**Outline of our compressed SGD analysis** The analysis above is not applicable to our algorithm due to gradient compression and error-feedback. Moreover, in the case of an arbitrary compressor we change the algorithm even further by periodically setting $\mathbf{e}_t$ to 0.

One of the major changes is that errors introduced by the compression lead to even greater deviation of SGD from the power method, and this deviation can potentially dominate other terms: if the compression error is accumulated from the beginning of the algorithm execution, then the compression error can be arbitrarily large. Let $\mathbf{e}'_t$ be the compression error sequence corresponding to $\mathbf{x}'_t$ such that

$\mathbf{e}_0' = \mathbf{e}_0$. The deviation of SGD from the power method caused by compression can be expressed as:

$$\eta^2 \nabla^2 f(\mathbf{x}_0) \sum_{i=1}^{t-1} (I - \eta \nabla^2 f(\mathbf{x}_0))^{t-1-i} (\mathbf{e}_i - \mathbf{e}_i'). \quad \text{(Proposition B.12 and Lemma B.16)}$$

It remains to bound $\|\mathbf{e}_i - \mathbf{e}_i'\|$ for all $i$. For $G_t = \mathbf{e}_t + \nabla F(\mathbf{x}_t, \theta_t) + \xi_t$ (with $G_t'$ defined analogously), since $\mathbf{e}_{t+1} = G_t - \mathcal{C}(G_t)$, by linearity of $\mathcal{C}$ we have:

$$\begin{aligned}
\mathbb{E}\left[\|\mathbf{e}_{t+1} - \mathbf{e}_{t+1}'\|^2\right] &= \mathbb{E}\left[\|(G_t - G_t') - \mathcal{C}(G_t - G_t', \tilde{\theta}_t)\|^2\right] \\
&\leq (1 - \mu)\mathbb{E}\left[\|G_t - G_t'\|^2\right] \\
&= (1 - \mu)\mathbb{E}\left[\|(\mathbf{e}_t - \mathbf{e}_t') + (\nabla F(\mathbf{x}_t, \theta_t) - \nabla F(\mathbf{x}_t', \theta_t)) + (\xi_t - \xi_t')\|^2\right].
\end{aligned}$$

Since $\mathbf{e}_0 = \mathbf{e}_0'$, after telescoping, $\|\mathbf{e}_{t+1} - \mathbf{e}_{t+1}'\|$ can be bounded using $\|\nabla F(\mathbf{x}_i, \theta_i) - \nabla F(\mathbf{x}_i', \theta_i)\|$ and $\|\xi_i - \xi_i'\|$ for $i \in [0:t]$ (Lemma B.17). In other words, when escaping from a saddle point, the deviation can bounded based on gradients and noises encountered during escaping. Therefore it is comparable to other terms and can be bounded with an appropriate choice of $\eta$.

Unfortunately, for the arbitrary compressor case we don't have a good estimation on $\mathcal{E}_t$, since in general we don't have better bound on $\|\mathbf{e}_i - \mathbf{e}_i'\|$ than $\|\mathbf{e}_i\| + \|\mathbf{e}_i'\|$ (see proof of Lemma B.16). Lemma A.5 bounds the compression error $\mathbf{e}_t$ in terms of $\|\nabla f(\mathbf{x}_0)\|, \ldots, \|\nabla f(\mathbf{x}_t)\|$:

$$\mathbb{E}\left[\|\mathbf{e}_t\|^2\right] \leq \frac{2(1 - \mu)}{\mu} \sum_{i=0}^{t-1} \left(1 - \frac{\mu}{2}\right)^{t-i} \mathbb{E}\left[\|\nabla f(\mathbf{x}_i)\|^2 + \chi^2\right],$$

but the bound depends on all gradients starting from the first iteration. To solve this problem, we periodically set the compression error to 0 (Line 5 of Algorithm 1). Let $t'$ be an iteration such that $\mathbf{e}_{t'}$ is set to 0: then, when escaping from $\mathbf{x}_{t'}$, we can apply Lemma A.5 with $i$ starting from $t'$. This leads to major difference from the [25] analysis: we need to consider large- and small-gradient cases separately. When the gradient at $\mathbf{x}_{t'}$ is large (Lemma B.7), we show that nearby gradients are also large, and the objective improves by the Compressed Descent Lemma A.7. Otherwise, we can bound the error norm for the next few iterations (Lemma B.16).

Finally, the analysis uses not only the sequence of iterates $\{\mathbf{x}_t\}$, but also the corrected sequence $\{\mathbf{y}_t\}$ where $\mathbf{y}_t = \mathbf{x}_t - \eta \mathbf{e}_t$ (similarly, $\mathbf{y}_t' = \mathbf{x}_t' - \eta \mathbf{e}_t'$). Intuitively, $\mathbf{e}_t$ accumulates the difference between the communicated and the original gradient, and therefore the goal of $\mathbf{y}_t$ is to offset the compression error. Typically, $\mathbf{x}_t$ is used as an argument of $\nabla f(\cdot)$, while $\mathbf{y}_t$ is used in distances and as an argument of $f(\cdot)$, which noticeably complicates the analysis. In particular, if some property holds for $\mathbf{x}_t$, it doesn't necessarily hold for $\mathbf{y}_t$ and vice versa: for example, since $\mathbf{x}_t$ and $\mathbf{y}_t$ are not necessarily close, bound $\|\mathbf{y}_t - \mathbf{y}_t'\|$ doesn't in general imply bound on $\|\mathbf{x}_t - \mathbf{x}_t'\|$. However, in our analysis, we show that we can bound $\|\mathbf{x}_t - \mathbf{x}_t'\|$, which is required to bound $\|\nabla f(\mathbf{x}_t) - \nabla f(\mathbf{x}_t')\|$ in Lemma B.18.

## 4   Experiments

In our experiments, we show that noisy Compressed SGD achieves convergence comparable with full SGD and successfully escapes saddle points. We perform our first set of experiments on ResNet34 model trained using CIFAR-10 dataset with step size 0.1. We distribute the data across 10 machines, such that each machine contains data from a single class. We analyze convergence of compressed SGD with RANDOMK compressor when $100\%$, $10\%$, $1\%$ and $0.1\%$ random gradient coordinates are communicated. Figure 1 shows that SGD with RANDOMK with $10\%$ or $1\%$ of coordinates compression converges as fast as the full SGD, while requiring substantially smaller communication.

In our second set of experiments, we show that SGD indeed encounters saddle points and noise facilitates escaping from them. We compare uncompressed SGD, SGD with TOPK compressor ($0.1\%$ of coordinates), and SGD with RANDOMK compressor ($0.1\%$ of coordinates) on deep MNIST autoencoder[7]. In all settings, we compare their convergence rates with and without noise. Figure 2

---

[7]The encoder is defined using 3 convolutional layers with ReLU activation, with the following parameters: (channels=16, kernel=3, stride=2, padding=1), (channels=32, kernel=3, stride=2, padding=1) and (channels=64, kernel=7, stride=1, padding=0). The decoder is symmetrical.

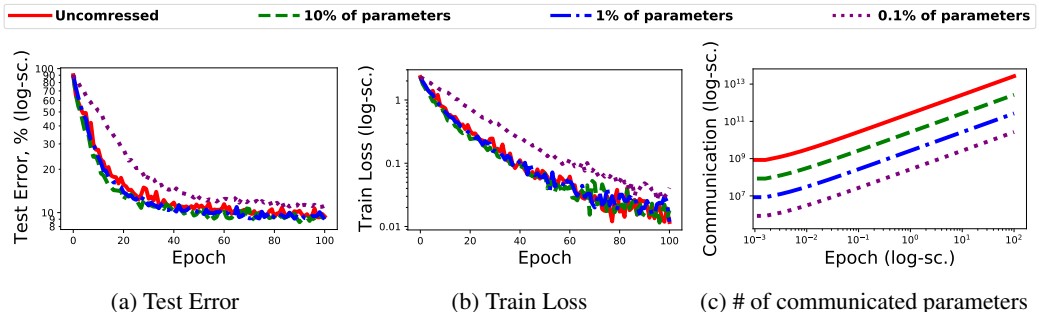

(a) Test Error          (b) Train Loss        (c) # of communicated parameters

Figure 1: Convergence of distributed SGD ($\eta = 0.1$, batch size is 8 per machine) with RANDOMK compressor when $100\%$ (full gradient), $10\%$, $1\%$ and $0.1\%$ of coordinates are used. ResNet34 model is trained on CIFAR-10 distributed across 10 machines: each machine corresponds to a single class. SGD with $10\%$ and $1\%$ compression achieves performance similar to that of uncompressed SGD, while requiring significantly less communication

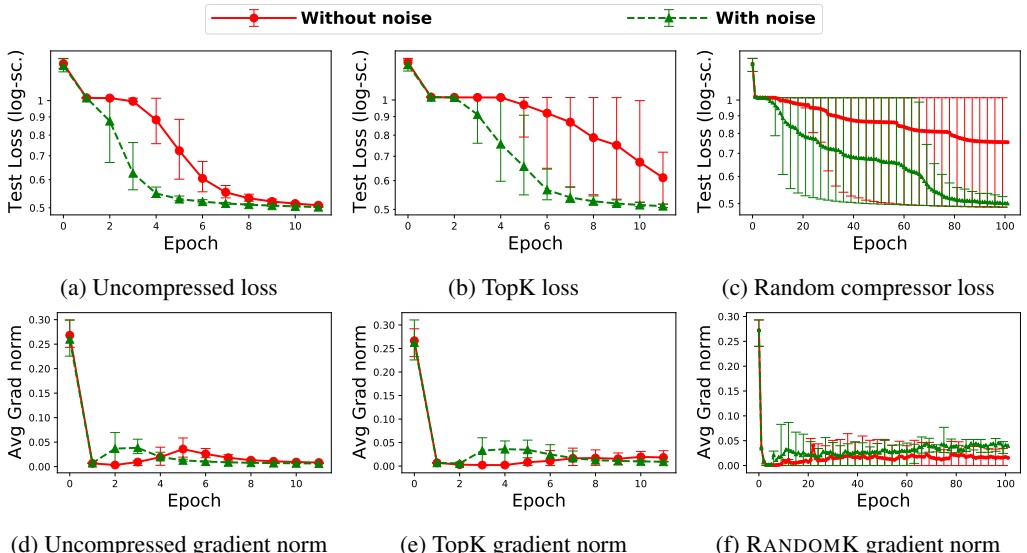

(a) Uncompressed loss      (b) TopK loss      (c) Random compressor loss

(d) Uncompressed gradient norm    (e) TopK gradient norm    (f) RANDOMK gradient norm

Figure 2: Convergence of SGD ($\eta = 0.1$, batch size is 64) without compression (left), with TOPK ($0.1\%$ of coordinates) compression (middle), and with RANDOMK ($0.1\%$ of coordinates) compression (right) on MNIST autoencoder dataset without noise (red) and with Gaussian noise (green, $\sigma = 0.01$ for each coordinate). Data points correspond to average over 10 executions and error bars correspond to $10\%$- and $90\%$-quantiles. The bottom row shows the norms of the stochastic gradients averaged over the last 100 iterations. The figure shows that SGD encounters and escapes saddle points for all compressors, and adding noise facilitates escaping from the saddle points[†].

[†] For the sake of presentation, to ensure that gradient converges to 0, we decrease the magnitude of the artificial noise at later iterations. With a fixed noise magnitude, as our theory predicts, gradient norm converges if a smaller step size is used, but this requires significantly more iterations. Note that this modification only affects the gradient convergence as the objective converges even with fixed noise and a large step size.

shows that SGD does encounter saddle points: e.g. in Figure 2a, for SGD without noise, during epochs 1-3, the gradient norm is close to 0 and the objective value doesn't improve. However, compressed SGD escapes from the saddle points, and noise significantly improves the escaping rate.

## 5  Conclusion

We give the first result for convergence of compressed SGD to an $\varepsilon$-SOSP, and it's possible that the convergence rate and the total communication can be further improved. When Assumption C holds,

it is likely that the communication can be improved by an $\varepsilon^{-1/4}$ factor using techniques from [19], which achieve $\tilde{O}(\varepsilon^{-3.5})$ convergence rate under Assumption C. Using variance reduction techniques, which achieve $\tilde{O}(\varepsilon^{-3})$ convergence rate, we expect $\varepsilon^{-1/2}$ improvement. Finally, it remains open whether linearity of the compressor is required for Theorem 3.2: similarly to the stochastic gradient case, it may suffice for the compressor to be Lipschitz.

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
