** function $\mathcal{C}(\mathbf{x}) = \frac{\|x\|_1}{d} sign(x)$ is a $1/d$-compressor [Bernstein et al., 2018]. Representation of $\mathcal{C}(x)$ requires $O(d)$ bits, but it is hard to compute in distributed settings: it's not clear how to find the signs of the coordinates without knowing the full vector, which requires each worker to send all coordinates. A practical solution is for each worker to communicate SIGN of its local gradient, and the final sign for each coordinate is selected by majority vote. Unfortunately, the resulting vector is not necessarily a compression of the gradient.

**QUANTIZATION** [Alistarh et al., 2017] uniformly splits segment $[0, \|\mathbf{x}\|]$ into $s$ buckets of the same size. Let $\ell_i = \left\lfloor \frac{|\mathbf{x}_i|}{\|\mathbf{x}\|/s} \right\rfloor$; then $|\mathbf{x}_i|$ is randomly rounded to one of $\ell_i\frac{\|\mathbf{x}\|}{s}$ and $(\ell_i + 1)\frac{\|\mathbf{x}\|}{s}$. The compressor returns non-zero coordinates after rounding. For $s = 1$, QUANTIZATION $Q(\mathbf{x})$ can be represented using $\tilde{O}(\sqrt{d})$ bits. While it doesn't fall into the compression framework, since $\|Q(\mathbf{x}) - \mathbf{x}\|$ can be much greater than $\|\mathbf{x}\|$, it has a property $\mathbb{E}[Q(\mathbf{x})] = \mathbf{x}$, which allows one to show convergence.

**TOPK** function preserves only $k$ largest (by the absolute value) coordinates of a vector and is a $k/d$-compressor [Stich et al., 2018]. This compressor can be represented using $\tilde{O}(k)$ bits, but similarly to SIGN, it is hard to compute in distributed settings. To address this issue, Alistarh et al. [2018] assume that TOPK of the average gradient is close to the average of TOPK of local gradients and show that this assumption holds in practice.

**Sketch-based TOPK** [Ivkin et al., 2019] is randomized communication-efficient compressor based on Count Sketch, which recovers top-$k$ coordinates in a distributed setting. It uses the fact that Count Sketch is a linear sketch (and therefore it can be easily combined across multiple machines) and can be used to recover top-$k$ coordinates of the vector with high probability. Therefore, it can be used as an efficient $k/d$-compressor requiring $\tilde{O}(k)$ communication.

**RANDOMK** compressor preserves $k$ random coordinates of a vector. It is a $k/d$-compressor [Stich et al., 2018] requiring $O(k)$ communication.

While it is known that compressed SGD converges (e.g. Karimireddy et al. [2019] and the works above), the convergence was shown only to a FOSP. The crucial idea to facilitate convergence is to use error-feedback [Stich et al., 2018]: the difference between the true gradient and its compression is propagated to the next iteration.

## 1.2   Our Contributions

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

 first result is similar to that of Stich et al. [2018] (after reformulation in terms of $\varepsilon$-FOSP), but is more general: it covers the case when $\mu$ is close to 0 and doesn't require any bounds on $\|\nabla F(\mathbf{x}, \theta)\|$ or $\|\nabla f_i(\mathbf{x}) - \nabla f(\mathbf{x})\|$, which are common assumptions in the literature (see Section 1.1). The proof of the theorem is presented in Appendix A.

---

**Theorem 3.1 (Convergence to $\varepsilon$-FOSP)** *Let $f$ satisfy Assumptions A and B and let $\mathcal{C}$ be a $\mu$-compressor. Then for Algorithm 1 with $reset\_error = false$ and $\eta = \tilde{O}\left(\min\left(\varepsilon^2, \frac{\mu}{\sqrt{1-\mu}}\varepsilon\right)\right)$, after $T = \tilde{O}(\frac{1}{\varepsilon^2\eta}) = \tilde{O}\left(\frac{1}{\varepsilon^4} + \frac{\sqrt{1-\mu}}{\mu\varepsilon^3}\right)$ iterations, at least half of visited points are $\varepsilon$-FOSP.*

---

**Corollary 3.2** *For a $1/d$-compressor with $\tilde{O}(1)$ communication (polylogarithmic on all parameters), the total communication per worker is $\tilde{O}\left(\frac{1}{\varepsilon^4} + \frac{d}{\varepsilon^3}\right)$, which outperforms full SGD communication $\tilde{O}\left(\frac{d}{\varepsilon^4}\right)$ by a factor of $\min\left(d, \varepsilon^{-1}\right)$.*

## 3.2 Convergence to an $\varepsilon$-SOSP

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

# A  Convergence to $\varepsilon$-FOSP

In this section we prove Theorem 3.1, showing that Algorithm 1 converges to an approximate first-order stationary point. Results and proofs are inspired by Karimireddy et al. [2019], with the key difference in that we show how to avoid using the bounded gradient assumption: $\mathbb{E}\left[\|\nabla F\|^2\right] \le G^2$ and handle the case of $\mu$-compressors with $\mu \ll 1$. Furthermore, Compressed Descent Lemma (Lemma A.7) is a foundation for showing a second-order convergence.

---

**Definition A.1 (Noise and compression parameters)** *We use the following notation:*
- $\zeta_t = \nabla F(\mathbf{x}_t, \theta_t) - \nabla f(\mathbf{x}_t)$ *is stochastic gradient noise. This noise has variance* $\sigma^2$
- $\xi_t$ *is artificial Gaussian noise added at every iteration. This noise has variance* $r^2$
- $\psi_t = \zeta_t + \xi_t$ *is the total noise. This noise has variance* $\chi^2 = \sigma^2 + r^2$.
- *We assume that gradients are compressed using a $\mu$-compressor $\mathcal{C}$.*

---

For the sake of the analysis, similarly to Karimireddy et al. [2019], we introduce an auxiliary sequence of corrected iterates $\{\mathbf{y}_t\}$, which remove the impact of the compression error.

---

**Definition A.2 (Corrected iterates)** *The sequence of corrected iterates $\{\mathbf{y}_t\}$ is defined as*

$$\mathbf{y}_t = \mathbf{x}_t - \eta \mathbf{e}_t$$

---

**Proposition A.3** *For the sequence $\{\mathbf{y}_t\}$, we have $\mathbf{y}_{t+1} - \mathbf{y}_t = -\eta(\nabla f(\mathbf{x}_t) + \psi_t)$*

**Proof :**    Recall that $\mathbf{e}_{t+1} = \nabla f(\mathbf{x}_t) + \psi_t + \mathbf{e}_t - \mathbf{g}_t$ and $\mathbf{g}_t = \mathcal{C}(\nabla f(\mathbf{x}_t) + \psi_t + \mathbf{e}_t, \theta_t)$ and thus

$$\mathcal{C}(\nabla f(\mathbf{x}_t) + \psi_t + \mathbf{e}_t, \tilde{\theta}_t) = \nabla f(\mathbf{x}_t) + \psi_t + \mathbf{e}_t - \mathbf{e}_{t+1}.$$

Substituting this into equation for $\mathbf{y}_{t+1}$:

$$
\begin{aligned}
\mathbf{y}_{t+1} &= \mathbf{x}_{t+1} - \eta \mathbf{e}_{t+1} \\
&= \mathbf{x}_t - \eta \mathcal{C}(\nabla f(\mathbf{x}_t) + \psi_t + \mathbf{e}_t, \tilde{\theta}_t) - \eta \mathbf{e}_{t+1} && (\text{Since } \mathbf{x}_{t+1} = \mathbf{x}_t - \eta \mathcal{C}(\nabla f(\mathbf{x}_t) + \psi_t + \mathbf{e}_t, \tilde{\theta}_t)) \\
&= \mathbf{x}_t - \eta(\nabla f(\mathbf{x}_t) + \psi_t + \mathbf{e}_t - \mathbf{e}_{t+1}) - \eta \mathbf{e}_{t+1} \\
&= \mathbf{x}_t - \eta(\nabla f(\mathbf{x}_t) + \psi_t + \mathbf{e}_t) \\
&= \mathbf{x}_t - \eta \mathbf{e}_t - \eta(\nabla f(\mathbf{x}_t) + \psi_t) \\
&= \mathbf{y}_t - \eta(\nabla f(\mathbf{x}_t) + \psi_t)
\end{aligned}
$$

$\square$

In our derivations, we'll often use conditional expectation with respect to current iterates.

---

**Definition A.4 (Conditional expectation w.r.t. iterate)** *Let $t$ be an iteration and $\xi$ be a random variable. Then*

$$\mathbb{E}_t\left[\xi\right] := \mathbb{E}_{(\theta_t, \tilde{\theta}_t, \xi_t), (\theta_{t+1}, \tilde{\theta}_{t+1}, \xi_{t+1}), \ldots}\left[\xi \mid \mathbf{x}_t, \mathbf{e}_t\right]$$

---

## A.1  Compression Error Bound

Recall that the compression error terms $\mathbf{e}_t$ in Algorithm 1 represent the difference between the computed gradient and the compressed gradient. Similarly to how stochastic noise increases the number of iterations compared with deterministic gradient descent, compression errors also increase the number of iterations, and therefore it's important to bound $\|\mathbf{e}_t\|$.

---

**Lemma A.5 (Compression Error Bound)** *Let* $\mathbf{x}_t, \mathbf{e}_t$ *be defined as in Algorithm 1 and let* $\chi^2$ *be as in Definition A.1. Then under Assumptions A and B, for any $t$ we have*

$$\mathbb{E}\left[\|\mathbf{e}_t\|^2\right] \leq \frac{2(1-\mu)}{\mu} \sum_{i=0}^{t-1}\left(1-\frac{\mu}{2}\right)^{t-i} \mathbb{E}\left[\|\nabla f(\mathbf{x}_i)\|^2 + \chi^2\right],$$

---

In particular, by considering a uniform bound on $\mathbb{E}\left[\|\nabla f(\mathbf{x}_i)\|^2\right]$ and taking the sum of the geometric series, we get a result similar to Karimireddy et al. [2019, Lemma 3]:

$$\mathbb{E}\left[\|\mathbf{e}_t\|^2\right] \leq \frac{4(1-\mu)}{\mu^2}\left(\max_{i=0}^{t-1} \mathbb{E}\left[\|\nabla f(\mathbf{x}_i)\|^2\right] + \chi^2\right)$$

**Proof :**   The proof is similar to the one of Karimireddy et al. [2019, Lemma 3]. The main difference is that we don't rely on the bounded gradient assumption.

By definition of $\mathbf{e}_{t+1}$:

$$\mathbb{E}\left[\|\mathbf{e}_{t+1}\|^2\right] = \mathbb{E}\left[\|\mathbf{e}_t + \nabla f(\mathbf{x}_t) + \psi_t - \mathcal{C}(\mathbf{e}_t + \nabla f(\mathbf{x}_t) + \psi_t, \theta_t)\|^2\right]$$
$$\leq (1-\mu)\mathbb{E}\left[\|\mathbf{e}_t + \nabla f(\mathbf{x}_t) + \psi_t\|^2\right]$$

By using inequality $\|a+b\|^2 \leq (1+\nu)\|a\|^2 + (1+\frac{1}{\nu})\|b\|^2$ for any $\nu$:

$$\mathbb{E}\left[\|\mathbf{e}_{t+1}\|^2\right] \leq (1-\mu)((1+\nu)\mathbb{E}\left[\|\mathbf{e}_t\|^2\right] + (1+\frac{1}{\nu})\mathbb{E}\left[\|\nabla f(\mathbf{x}_t) + \psi_t\|^2\right])$$

$$\leq \sum_{i=0}^{t}(1-\mu)^{t-i+1}(1+\nu)^{t-i}(1+\frac{1}{\nu})\mathbb{E}\left[\|\nabla f(\mathbf{x}_i) + \psi_i\|^2\right] \qquad \text{(Telescoping)}$$

$$\leq \frac{1}{\nu}\sum_{i=0}^{t}((1-\mu)(1+\nu))^{t-i+1}\mathbb{E}\left[\|\nabla f(\mathbf{x}_i) + \psi_i\|^2\right]$$

By selecting $\nu = \frac{\mu}{2(1-\mu)}$, we have $(1-\mu)(1+\nu) = 1 - \frac{\mu}{2}$. Therefore:

$$\mathbb{E}\left[\|\mathbf{e}_{t+1}\|^2\right] \leq \frac{2(1-\mu)}{\mu}\sum_{i=0}^{t}\left(1-\frac{\mu}{2}\right)^{t-i+1}\mathbb{E}\left[\|\nabla f(\mathbf{x}_i) + \psi_i\|^2\right]$$

$$= \frac{2(1-\mu)}{\mu}\sum_{i=0}^{t}\left(1-\frac{\mu}{2}\right)^{t-i+1}\mathbb{E}\left[\|\nabla f(\mathbf{x}_i)\|^2 + \chi^2\right] \qquad (\mathbb{E}\left[\chi \mid \mathbf{x}_i\right] = 0)$$

$\square$

For the sum of $\|\mathbf{e}_t\|^2$, we have the following, simpler expression:

**Corollary A.6** *Under assumptions of Lemma A.5, we have*

$$\sum_{\tau=0}^{t} \mathbb{E}\left[\|\mathbf{e}_\tau\|^2\right] \leq \frac{4(1-\mu)}{\mu^2} \sum_{\tau=0}^{t} (\mathbb{E}\left[\|\nabla f(\mathbf{x}_\tau)\|^2\right] + \chi^2)$$

**Proof :**

$$\sum_{\tau=0}^{t} \mathbb{E}\left[\|\mathbf{e}_\tau\|^2\right] \leq \frac{2(1-\mu)}{\mu} \sum_{\tau=0}^{t} \sum_{i=0}^{\tau} \left(1 - \frac{\mu}{2}\right)^{\tau-i+1} \mathbb{E}\left[\|\nabla f(\mathbf{x}_i)\|^2 + \chi^2\right]$$

$$\leq \frac{2(1-\mu)}{\mu} \sum_{i=0}^{t} \left(\mathbb{E}\left[\|\nabla f(\mathbf{x}_i)\|^2 + \chi^2\right] \sum_{\tau=i}^{t} \left(1 - \frac{\mu}{2}\right)^{\tau-i+1}\right)$$

Bounding $\sum_\tau \left(1 - \frac{\mu}{2}\right)^{\tau-i+1}$ with the sum of the geometric series $\frac{2}{\mu}$, we have:

$$\sum_{\tau=0}^{t} \mathbb{E}\left[\|\mathbf{e}_\tau\|^2\right] \leq \frac{4(1-\mu)}{\mu^2} \sum_{i=0}^{t} \mathbb{E}\left[\|\nabla f(\mathbf{x}_i)\|^2 + \chi^2\right]$$

$\square$

## A.2 Compressed Descent Lemma

The following descent lemma is the key tool in the analysis as it allows us to bound gradient norms across multiple iterations.

---

**Lemma A.7 (Compressed Descent Lemma)** *Let $f$ satisfy Assumptions A and B and $\chi^2$ be as in Definition A.1. For $\eta < \frac{1}{4L} \min(\frac{\mu}{\sqrt{1-\mu}}, 1)$, for any $T$ we have:*

$$\sum_{\tau=0}^{T-1} \mathbb{E}\left[\|\nabla f(\mathbf{x}_\tau)\|^2\right] \leq \frac{4(f(\mathbf{y}_0) - \mathbb{E}\left[f(\mathbf{y}_T)\right])}{\eta} + \eta T \chi^2 \left(2L + \frac{8L^2\eta(1-\mu)}{\mu^2}\right)$$

---

Using this lemma, we'll later show that for sufficiently large $T$, multiple visited points have small gradients (note that by dividing the left-hand side by $T$ we obtain an average squared gradient norm), making them $\varepsilon$-FOSP. On the right-hand side the first term is bounded by $4f_{\max}/\eta$, while the other two terms can be bounded by selecting a sufficiently small $\eta$. The second term arises from stochastic gradient noise, while the last term stems from the compression error.

**Proof :** The proof is similar to the one of Karimireddy et al. [2019, Theorem II]. By the folklore descent lemma, using notation $\mathbb{E}_t\left[\cdot\right]$ from Definition A.4:

$\mathbb{E}_t\left[f(\mathbf{y}_{t+1})\right]$

$\leq f(\mathbf{y}_t) + \langle \nabla f(\mathbf{y}_t), \mathbb{E}_t\left[\mathbf{y}_{t+1} - \mathbf{y}_t\right]\rangle + \frac{L}{2}\mathbb{E}_t\left[\|\mathbf{y}_{t+1} - \mathbf{y}_t\|^2\right]$

$= f(\mathbf{y}_t) - \eta \mathbb{E}_{\theta_t, \tilde{\theta}_t}\left[\langle \nabla f(\mathbf{y}_t), \nabla f(\mathbf{x}_t) + \psi_t\rangle \mid \mathbf{x}_t, \mathbf{e}_t\right] + \frac{L\eta^2}{2}\mathbb{E}_{\theta_t, \tilde{\theta}_t}\left[\|\nabla f(\mathbf{x}_t) + \psi_t\|^2 \mid \mathbf{x}_t, \mathbf{e}_t\right]$ *(Prop. A.3)*

$\leq f(\mathbf{y}_t) - \eta\|\nabla f(\mathbf{x}_t)\|^2 - \eta\langle \nabla f(\mathbf{y}_t) - \nabla f(\mathbf{x}_t), \nabla f(\mathbf{x}_t)\rangle + \frac{L\eta^2}{2}\|\nabla f(\mathbf{x}_t)\|^2 + \frac{L\eta^2\chi^2}{2}$ $(\mathbb{E}\left[\psi_t\right] = 0)$

$\leq f(\mathbf{y}_t) - \eta\left(1 - \frac{L\eta}{2}\right)\|\nabla f(\mathbf{x}_t)\|^2 + \frac{L\eta^2\chi^2}{2} - \eta\langle \nabla f(\mathbf{y}_t) - \nabla f(\mathbf{x}_t), \nabla f(\mathbf{x}_t)\rangle$

Using inequality $|\langle a, b \rangle| \leq \frac{\|a\|^2}{2} + \frac{\|b\|^2}{2}$ and smoothness, we have:

$$\mathbb{E}_t\left[f(\mathbf{y}_{t+1}) \mid \mathbf{x}_t, \mathbf{e}_t\right]$$

$$\leq f(\mathbf{y}_t) - \eta\left(1 - \frac{L\eta}{2}\right)\|\nabla f(\mathbf{x}_t)\|^2 + \frac{L\eta^2\chi^2}{2} + \frac{\eta}{2}\|\nabla f(\mathbf{y}_t) - \nabla f(\mathbf{x}_t)\|^2 + \frac{\eta}{2}\|\nabla f(\mathbf{x}_t)\|^2$$

$$\leq f(\mathbf{y}_t) - \eta\left(\frac{1}{2} - \frac{L\eta}{2}\right)\|\nabla f(\mathbf{x}_t)\|^2 + \frac{L\eta^2\chi^2}{2} + \frac{\eta L^2}{2}\|\mathbf{y}_t - \mathbf{x}_t\|^2 \qquad (L\text{-smoothness})$$

$$\leq f(\mathbf{y}_t) - \eta\left(\frac{1}{2} - \frac{L\eta}{2}\right)\|\nabla f(\mathbf{x}_t)\|^2 + \frac{L\eta^2\chi^2}{2} + \frac{\eta^3 L^2}{2}\|\mathbf{e}_t\|^2 \qquad (\text{Def. A.2 of } \mathbf{y}_t)$$

Using telescoping and taking the expectation, we bound $f(\mathbf{y}_{t+1})$:

$$\mathbb{E}\left[f(\mathbf{y}_{t+1})\right] \leq f(\mathbf{y}_0) - \eta\left(\frac{1}{2} - \frac{L\eta}{2}\right)\sum_{\tau=0}^{t}\mathbb{E}\left[\|\nabla f(\mathbf{x}_\tau)\|^2\right] + \frac{L\chi^2\eta^2(t+1)}{2} + \eta^3 L^2 \sum_{\tau=0}^{t}\mathbb{E}\left[\|\mathbf{e}_\tau\|^2\right]$$

Bounding $\sum_\tau \|\mathbf{e}_\tau\|^2$ by Corollary A.6, we have:

$$\mathbb{E}\left[f(\mathbf{y}_t)\right]$$

$$\leq f(\mathbf{y}_0) - \eta\left(\frac{1}{2} - \frac{L\eta}{2}\right)\sum_{\tau=0}^{t-1}\mathbb{E}\left[\|\nabla f(\mathbf{x}_\tau)\|^2\right] + \frac{L\chi^2\eta^2 t}{2} + \frac{2\eta^3 L^2(1-\mu)}{\mu^2}\sum_{i=0}^{t-1}\mathbb{E}\left[\|\nabla f(\mathbf{x}_i)\|^2 + \chi^2\right]$$

$$\leq f(\mathbf{y}_0) - \eta\left(\frac{1}{2} - \frac{L\eta}{2} - \frac{2\eta^2 L^2(1-\mu)}{\mu^2}\right)\sum_{\tau=0}^{t-1}\mathbb{E}\left[\|\nabla f(\mathbf{x}_\tau)\|^2\right] + \frac{L\chi^2\eta^2 t}{2} + \frac{2\eta^3 L^2\chi^2(1-\mu)t}{\mu^2}$$

Using that $\eta < \frac{1}{4L}\min\left(\frac{\mu}{\sqrt{1-\mu}}, 1\right)$, we bound the coefficient before $\sum_{\tau=0}^{t}\mathbb{E}\left[\|\nabla f(\mathbf{x}_\tau)\|^2\right]$ with $\frac{\eta}{4}$:

$$\mathbb{E}\left[f(\mathbf{y}_t)\right] \leq f(\mathbf{y}_0) - \frac{\eta}{4}\sum_{\tau=0}^{t-1}\mathbb{E}\left[\|\nabla f(\mathbf{x}_\tau)\|^2\right] + \eta^2\chi^2 t\left(\frac{L}{2} + \frac{2L^2\eta(1-\mu)}{\mu^2}\right)$$

After regrouping the terms, we get the final result:

$$\sum_{\tau=0}^{t-1}\mathbb{E}\left[\|\nabla f(\mathbf{x}_\tau)\|^2\right] \leq \frac{4(f(\mathbf{y}_0) - \mathbb{E}\left[f(\mathbf{y}_t)\right])}{\eta} + \eta\chi^2 t\left(2L + \frac{8L^2\eta(1-\mu)}{\mu^2}\right)$$

$\square$

When showing convergence to SOSP, we'll need a generalization of this Lemma which start tracking communication error from the last iteration when the error was 0:

**Corollary A.8** *Let $f$ satisfy Assumptions A and B and $\chi^2$ be as in Definition A.1. If $t_0$ is an iteration of Algorithm 1 such that $\mathbf{e}_{t_0} = 0$ and $\eta < \frac{1}{4L}\min(\frac{\mu}{\sqrt{1-\mu}}, 1)$, then for any $T$ we have:*

$$\sum_{\tau=0}^{T-1}\mathbb{E}\left[\|\nabla f(\mathbf{x}_{t_0+\tau})\|^2\right] \leq \frac{4(f(\mathbf{y}_{t_0}) - \mathbb{E}\left[f(\mathbf{y}_{t_0+T})\right])}{\eta} + \eta T\chi^2\left(2L + \frac{8L^2\eta(1-\mu)}{\mu^2}\right)$$

## A.3 Convergence to $\varepsilon$-FOSP

> **Theorem A.9 (Convergence to $\varepsilon$-FOSP)** *Let $f$ satisfy Assumptions A and B. Then for $\eta = \tilde{O}\left(\min\left(\varepsilon^2, \frac{\mu}{\sqrt{1-\mu}}\varepsilon\right)\right)$, after $T = \tilde{\Theta}\left(\frac{1}{\varepsilon^4} + \frac{\sqrt{1-\mu}}{\mu\varepsilon^3}\right)$ iterations, at least half of visited points are $\varepsilon$-FOSP.*

**Proof :** Proof by contradiction. For $\eta < \frac{1}{4L}\min\left(\frac{\mu}{\sqrt{1-\mu}}, 1\right)$, if less than half points are $\varepsilon$-FOSP, then by Lemma A.7:

$$\frac{T\varepsilon^2}{2} \leq \sum_{\tau=0}^{T} \mathbb{E}\left[\|\nabla f(\mathbf{x}_\tau)\|^2\right] \leq \frac{4f_{\max}}{\eta} + \eta\chi^2 T\left(2L + \frac{8L^2\eta(1-\mu)}{\mu^2}\right)$$

It suffices to guarantee that all terms on the right-hand side are at most $\frac{T\varepsilon^2}{6}$:

$$2L\eta\chi^2 T \leq \frac{T\varepsilon^2}{6} \iff \eta \leq \frac{\varepsilon^2}{12L\chi^2} \qquad\qquad = \tilde{\Theta}(\varepsilon^2)$$

$$\frac{8L^2\chi^2\eta^2 T(1-\mu)}{\mu^2} \leq \frac{T\varepsilon^2}{6} \iff \eta \leq \frac{\mu\varepsilon}{\sqrt{1-\mu}L\chi\sqrt{48}} \qquad = \tilde{\Theta}\left(\frac{\mu\varepsilon}{\sqrt{1-\mu}}\right)$$

$$\frac{4f_{\max}}{\eta} \leq \frac{T\varepsilon^2}{6} \iff T \geq \frac{24f_{\max}}{\varepsilon^2\eta} \qquad\qquad = \tilde{\Theta}\left(\frac{1}{\eta\varepsilon^2}\right) = \tilde{\Theta}\left(\frac{1}{\varepsilon^4} + \frac{\sqrt{1-\mu}}{\mu\varepsilon^3}\right)$$

Therefore, after $\tilde{\Theta}\left(\frac{1}{\varepsilon^4} + \frac{\sqrt{1-\mu}}{\mu\varepsilon^3}\right)$ iterations at least half of the points are $\varepsilon$-FOSP. $\qquad\square$

# B    Convergence to $\varepsilon$-SOSP

By rescaling we can assume that $\varepsilon \leq 1$. Recall that $\alpha = 1$ when Assumption C holds and $\alpha = d$ otherwise. We introduce the following auxiliary notation:

---

**Definition B.1 (Step sizes)**

$$\text{max } \eta \text{ for SGD} \quad \eta_\sigma = \frac{\varepsilon^2}{L(1 + d\sigma^2)} + \min\left(\frac{\varepsilon^2}{L(1 + \sigma^2)}, \frac{\sqrt{\rho\varepsilon}}{\tilde{\ell}^2}\right) = \tilde{O}\left(\frac{\varepsilon^2}{\alpha}\right)$$

max $\eta$ for compressed SGD:

$$\text{For a general compressor:} \quad \eta_\mu = \min\left(\frac{\mu\varepsilon}{\sqrt{1-\mu}L\sigma}, \frac{\mu^2\sqrt{\varepsilon}}{(1-\mu)L^2 d}\right) = \tilde{O}\left(\min\left(\frac{\mu\varepsilon}{\sqrt{1-\mu}}, \frac{\mu^2\sqrt{\varepsilon}}{(1-\mu)d}\right)\right)$$

$$\text{For a linear compressor:} \quad \eta_\mu = \min\left(\frac{\mu\varepsilon}{\sqrt{1-\mu}L\sigma}, \frac{\mu^2\sqrt{\rho\varepsilon}}{(1-\mu)L^2}\right) = \tilde{O}\left(\min\left(\frac{\mu\varepsilon}{\sqrt{1-\mu}}, \frac{\mu^2\sqrt{\varepsilon}}{1-\mu}\right)\right)$$

---

Intuitively, selecting step size $\eta \leq \eta_\sigma$ suffices to show convergence of SGD [Jin et al., 2021]. In addition, selecting $\eta \leq \eta_\mu$ allows us to extend the results to compressed SGD.

---

**Definition B.2** *Our choice of parameters is the following ($c_\eta, c_\mathcal{I}, c_\mathcal{R}, c_\mathcal{F}, c_r$ hide polylogarithmic dependence on all parameters, the conditions on them will be specified later):*

$$\begin{aligned}
\text{Step size} \quad & \eta = c_\eta \min(\eta_\sigma, \eta_\mu) \\
\text{Iterations required for escaping} \quad & \mathcal{I} = c_\mathcal{I} \frac{1}{\eta\sqrt{\rho\varepsilon}} \\
\text{Escaping radius} \quad & \mathcal{R} = c_\mathcal{R}\sqrt{\frac{\varepsilon}{\rho}} \\
\text{Objective change after escaping} \quad & \mathcal{F} = c_\mathcal{F}\sqrt{\frac{\varepsilon^3}{\rho}} \\
\text{Noise standard deviation} \quad & r = c_r \frac{\varepsilon}{\sqrt{L\eta}}
\end{aligned} \tag{1}$$

---

Recall that $\chi^2 = \sigma^2 + r^2 = \sigma^2 + \frac{c_r\varepsilon^2}{L\eta}$ by Definition A.1 and $f_{\max} = f(\mathbf{x}_{t_0}) - f(\mathbf{x}^*)$. We will show that after $\mathcal{I}$ iterations the objective decreases by $\mathcal{F}$. Therefore, the objective decreases on average by $\frac{\mathcal{F}}{\mathcal{I}} = \tilde{\Omega}(\varepsilon^2\eta)$ per iteration resulting in $\tilde{O}\left(\frac{f_{\max}}{\varepsilon^2\eta}\right)$ iterations overall. See Table 1 for the number of iterations and total communication in various settings.

Intuitively, the motivation for this choice of parameters is the following. Let $\mathbf{x}$ be a point such that $\lambda_{\min}(\nabla^2 f(\mathbf{x})) < -\sqrt{\rho\varepsilon}$ and $\|\nabla f(\mathbf{x})\| = 0$.

- Our analysis happens inside $B(\mathbf{x}, \mathcal{R})$, and we want $\lambda_{\min}(\nabla^2 f(\mathbf{z})) < -\frac{\sqrt{\rho\varepsilon}}{2}$ for all $\mathbf{z} \in B(\mathbf{x}, \mathcal{R})$. By the Hessian-Lipschitz property, for $\mathbf{z} \in B(\mathbf{x}, \mathcal{R})$ we have $\|\nabla^2 f(\mathbf{x}) - \nabla^2 f(\mathbf{z})\| \leq \rho\mathcal{R}$. To have $\rho\mathcal{R} \leq \frac{\sqrt{\rho\varepsilon}}{2}$, we choose $\mathcal{R} \leq \frac{1}{2}\sqrt{\frac{\varepsilon}{\rho}}$.

- Let $-\gamma$ be the smallest eigenvalue and $\mathbf{v}_1$ be the corresponding eigenvector of $\nabla f^2(\mathbf{x})$. Assume that our function is quadratic and, after adding noise, the projection on $\mathbf{v}_1$ is $\tilde{\Theta}(1)$ (it is actually polynomial or reverse-polynomial on all parameters, which doesn't change the idea).

Table 2: Convergence to $\varepsilon$-SOSP for full SGD and for constant-size compression (the choice of parameters is not optimal; see Table 1 for the optimal choice). For any choice of $\mu$ and $\eta$: $T = \tilde{O}(1/\eta\varepsilon^2)$, $\mathcal{R} = \tilde{O}(\sqrt{\varepsilon})$, $\mathcal{F} = \tilde{O}(\sqrt{\varepsilon^3})$.

| Settings | $\mu$ | $\eta$ | $\mathcal{I}$ | $r$ |
|---|---|---|---|---|
| Uncompressed Lipschitz $\nabla F$ | $0$ | $\tilde{O}\left(\varepsilon^2\right)$ | $\tilde{O}\left(\varepsilon^{3/2}\right)$ | $\tilde{O}\left(1\right)$ |
| Compressed Lipschitz $\nabla F$ | $\frac{1}{d}$ | $\tilde{O}\left(\min\left(\varepsilon^2, \frac{\varepsilon}{d}, \frac{\sqrt{\varepsilon}}{d^3}\right)\right)$ | $\tilde{O}\left(\frac{1}{\eta\sqrt{\varepsilon}}\right)$ | $\tilde{O}\left(\frac{\varepsilon}{\sqrt{\eta}}\right)$ |
| RandomK Lipschitz $\nabla F$ | $\frac{1}{d}$ | $\tilde{O}\left(\min\left(\varepsilon^2, \frac{\varepsilon}{d}, \frac{\sqrt{\varepsilon}}{d^2}\right)\right)$ | $\tilde{O}\left(\frac{1}{\eta\sqrt{\varepsilon}}\right)$ | $\tilde{O}\left(\frac{\varepsilon}{\sqrt{\eta}}\right)$ |
| Uncompressed non-Lipschitz $\nabla F$ | $0$ | $\tilde{O}\left(\frac{\varepsilon^2}{d}\right)$ | $\tilde{O}\left(d\varepsilon^{3/2}\right)$ | $\tilde{O}\left(\sqrt{d}\right)$ |
| Compressed non-Lipschitz $\nabla F$ | $\frac{1}{d}$ | $\tilde{O}\left(\min\left(\frac{\varepsilon^2}{d}, \frac{\sqrt{\varepsilon}}{d^3}\right)\right)$ | $\tilde{O}\left(\frac{1}{\eta\sqrt{\varepsilon}}\right)$ | $\tilde{O}\left(\frac{\varepsilon}{\sqrt{\eta}}\right)$ |
| RandomK non-Lipschitz $\nabla F$ | $\frac{1}{d}$ | $\tilde{O}\left(\min\left(\frac{\varepsilon^2}{d}, \frac{\sqrt{\varepsilon}}{d^2}\right)\right)$ | $\tilde{O}\left(\frac{1}{\eta\sqrt{\varepsilon}}\right)$ | $\tilde{O}\left(\frac{\varepsilon}{\sqrt{\eta}}\right)$ |

Then after $t$ iterations, this projection increases by the factor of $(1 + \eta\gamma)^t$. For every $1/\eta\gamma$ iterations, the projection increases approximately by the factor of $e$. Therefore, to reach $\mathcal{R}$ starting from $\Theta(1)$, we need $\tilde{O}(\frac{1}{\eta\gamma})$ iterations, which is at most $\tilde{O}(\frac{1}{\eta\sqrt{\rho\varepsilon}})$

- In a certain sense, the best improvement we can hope to achieve is by moving from $\mathbf{x}$ to $\mathbf{x} + \mathcal{R}\mathbf{v}_1$. If $\nabla f(\mathbf{x}) = 0$ and the objective is quadratic in direction $\mathbf{v}_1$ with eigenvalue $\gamma$, the objective decreases by $\gamma\mathcal{R}^2 = \Omega(\sqrt{\frac{\varepsilon^3}{\rho}})$, which motivates the choice of $\mathcal{F}$.

- Bound on $r$ arises from the fact that $\chi^2 \approx r^2$ and that we want to bound the last term in Lemma B.4 with $\mathcal{F}$.

We formalize the first item in the following proposition:

**Proposition B.3** *Let $\mathbf{x}$ be a point such that $\lambda_{\min}(\nabla^2 f(\mathbf{x})) < -\sqrt{\rho\varepsilon}$. Then for any $\mathbf{z} \in B(\mathbf{x}, \mathcal{R})$, $\lambda_{\min}(\nabla^2 f(\mathbf{z})) < -\sqrt{\rho\varepsilon}/2$.*

## B.1 Proof outline

Our proof is mainly based on the ideas from Jin et al. [2021]. We first introduce "Improve or localize" lemma (Lemma B.4): if after the limited number of iterations the objective doesn't sufficiently improve, we conclude that we didn't move far from the original point. Similarly to Jin et al. [2021], we introduce a notion of coupling sequences: two gradient descent sequences having the same distribution such that, as long as we start from a saddle point, at least one of these sequences escapes, and therefore its objective improves. Since distributions of these sequences match distribution of sequence generated by gradient descent, we conclude that the algorithm sufficiently improves the objective.

Our analysis differs from Jin et al. [2021] in several ways. The first difference is that, aside from $\{\mathbf{x}_{t_0+t}\}$, our equations use another sequence $\{\mathbf{y}_{t_0+t}\}$ ($\mathbf{x}_{t_0+t}$ mainly participate as arguments of $\nabla f(\cdot)$, while $\mathbf{y}_{t_0+t}$ participate as argument of $f(\cdot)$ and in distances). This leads to the following challenge: if some relation holds for $\mathbf{y}_{t_0+t}$, it doesn't necessary holds for $\mathbf{x}_{t_0+t}$. For example, if we

have a bound on $\|\mathbf{y}_{t_0+t} - \mathbf{y}'_{t_0+t}\|$, we don't necessarily have a bound on $\|\mathbf{x}_{t_0+t} - \mathbf{x}'_{t_0+t}\|$, and it needs to be established separately.

Another difference is that, for a general compressor, we have to split our analysis into two parts: large gradient case and small gradient case. When our initial gradient is large, then we either escape the saddle points or the nearby gradients are also large, and by Lemma A.7 the objective improves (see details in Lemma B.7). If the gradient is small, we use "Improve or localize" Lemma as described above. In the latter case, similarly to Jin et al. [2021], we have to bound errors which arise from the fact that the function is not quadratic and gradients are not deterministic (see Definition B.11). However, we have an additional error term stemming from gradient compression (see Definition B.11); to bound this term (see Lemma B.16), we need bounded $\|\mathbf{e}_{t_0+t}\|$, and for that we use our assumptions that gradients are small.

## B.2 Improve or localize

We first show that, if gradient descent moves far enough from the initial point, then function value sufficiently decreases. The following lemma considers the general case, while Corollary B.5 considers the simplified form, obtained by substituting parameters from Definition B.2.

---

**Lemma B.4 (Improve or localize)** *Let $f$ satisfy Assumptions A and B and let $\mathbf{y}_{t_0+t}$ and $\chi$ be defined as in Definition A.1. If $t_0$ is an iteration of Algorithm 1 such that $\mathbf{e}_{t_0} = 0$, then using notation $\mathbb{E}_t[\cdot]$ from Definition A.4, for $\eta < \frac{1}{4L}\min(\frac{\mu}{\sqrt{1-\mu}}, 1)$ we have*

$$f(\mathbf{y}_{t_0}) - \mathbb{E}_{t_0}[f(\mathbf{y}_{t_0+t})] \geq \frac{\mathbb{E}_{t_0}\left[\|\mathbf{y}_{t_0+t} - \mathbf{y}_{t_0}\|^2\right]}{8\eta t} - \eta^2\chi^2 t\left(L + \frac{2(1-\mu)L^2\eta}{\mu^2}\right) - \eta\chi^2$$

---

**Proof :**     Let $\psi_t = \zeta_t + \xi_{t_0+t}$. By Proposition A.3, $\mathbf{y}_{i+1} = \mathbf{y}_{t_0+i} - \eta(\nabla f(\mathbf{x}_{t_0+i}) + \psi_i)$. Since noises are independent:

$$\mathbb{E}_{t_0}\left[\|\sum_{i=0}^{t-1}\psi_{t_0+i}\|^2\right] = \sum_{i=0}^{t-1}\mathbb{E}_{t_0}\left[\|\psi_{t_0+i}\|^2\right] = \sum_{\tau=0}^{t-1}\chi^2 = t\chi^2$$

By Proposition A.3:

$$\mathbb{E}_{t_0}\left[\|\mathbf{y}_{t_0+t} - \mathbf{y}_{t_0}\|^2\right] = \eta^2\mathbb{E}_{t_0}\left[\|\sum_{i=0}^{t-1}(\nabla f(\mathbf{x}_{t_0+i}) + \psi_{t_0+i})\|^2\right]$$

$$\leq 2\eta^2\mathbb{E}_{t_0}\left[\|\sum_{i=0}^{t-1}\nabla f(\mathbf{x}_{t_0+i})\|^2 + \|\sum_{i=0}^{t-1}\psi_{t_0+i}\|^2\right]$$

$$\leq 2\eta^2 t\sum_{i=0}^{t-1}\mathbb{E}_{t_0}\left[\|\nabla f(\mathbf{x}_{t_0+i})\|^2\right] + 2\eta^2\chi^2 t$$

Since $\eta < \frac{1}{4L}\min(\frac{\mu}{\sqrt{1-\mu}}, 1)$, by Corollary A.8:

$$\mathbb{E}_{t_0}\left[\|\mathbf{y}_{t_0+t} - \mathbf{y}_{t_0}\|^2\right] \leq 2\eta^2 t\left(\frac{4(f(\mathbf{y}_{t_0}) - \mathbb{E}_{t_0}[f(\mathbf{y}_{t_0+t})])}{\eta} + \eta\chi^2 t\left(2L + \frac{8(1-\mu)L^2\eta}{\mu^2}\right)\right) + 2\eta^2\chi^2 t$$

$$\leq 2\eta t\left(4(f(\mathbf{y}_{t_0}) - \mathbb{E}_{t_0}[f(\mathbf{y}_{t_0+t})]) + \eta^2\chi^2 t\left(4L + \frac{8(1-\mu)L^2\eta}{\mu^2}\right) + 4\eta\chi^2\right)$$

After regrouping the terms, we have:

$$f(\mathbf{y}_{t_0}) - \mathbb{E}_{t_0}\left[f(\mathbf{y}_{t_0+t})\right] \geq \frac{\mathbb{E}_{t_0}\left[\|\mathbf{y}_{t_0+t} - \mathbf{y}_{t_0}\|^2\right]}{8\eta t} - \eta^2\chi^2 t\left(L + \frac{2(1-\mu)L^2\eta}{\mu^2}\right) - \eta\chi^2,$$

$\square$

**Corollary B.5** *Let $t_0$ be an iteration of Algorithm 1 such that $\mathbf{e}_{t_0} = 0$. Under Assumptions A and B, for $\mathcal{F}, \mathcal{I}$ chosen as specified in Definition B.2, for any $t \leq \mathcal{I}$ we have:*

$$f(\mathbf{y}_{t_0}) - \mathbb{E}_{t_0}\left[f(\mathbf{y}_{t_0+t})\right] \geq \frac{\sqrt{\rho\varepsilon}}{8c_{\mathcal{I}}}\mathbb{E}_{t_0}\left[\|\mathbf{y}_{t_0+t} - \mathbf{y}_{t_0}\|^2\right] - \mathcal{F}$$

**Proof :**  By Lemma B.4, the first term on the right-hand side stems from $t \leq \mathcal{I} = \frac{c_{\mathcal{I}}}{\eta\sqrt{\rho\varepsilon}}$. With our choice of parameters, we can bound negative terms with $\mathcal{F}$ (recall that $\mathcal{F} = c_{\mathcal{F}}\sqrt{\frac{\varepsilon^3}{\rho}}$).

**Bounding $\eta\chi^2$.**

$$\eta\chi^2 = \eta\sigma^2 + \eta r^2 \leq c_\eta\frac{\varepsilon^2}{L} + c_r^2\frac{\varepsilon^2}{L} = (c_\eta + c_r^2)\frac{\sqrt{\varepsilon^3}}{\sqrt{\rho}} \cdot \frac{\sqrt{\rho\varepsilon}}{L} \leq (c_\eta + c_r^2)\frac{\sqrt{\varepsilon^3}}{\sqrt{\rho}},$$

where we use that $\sqrt{\rho\varepsilon} \leq L$, since otherwise all $\varepsilon$-FOSP are $\varepsilon$-SOSP.

**Bounding $\eta^2\chi^2 tL$.**  Since $\mathcal{I} = c_{\mathcal{I}}\frac{1}{\eta\sqrt{\rho\varepsilon}}$ and $t \leq \mathcal{I}$:

$$\eta^2\chi^2 tL \leq \frac{\eta\chi^2 L}{\sqrt{\rho\varepsilon}} \leq \eta\chi^2,$$

and we use the estimation above.

**Bounding $\eta^2\chi^2 t \cdot \frac{2(1-\mu)L^2\eta}{\mu^2}$.**

$$\begin{aligned}
\frac{\eta^3\chi^2 t(1-\mu)L^2}{\mu^2} &\leq \frac{c_{\mathcal{I}}\eta^2\chi^2(1-\mu)L^2}{\mu^2\sqrt{\rho\varepsilon}} && \left(t \leq \mathcal{I} = \frac{c_{\mathcal{I}}}{\eta\sqrt{\rho\varepsilon}}\right)\\
&\leq \frac{c_{\mathcal{I}}\eta^2(1-\mu)L^2\left(\sigma^2 + \frac{c_r\varepsilon^2}{L\eta}\right)}{\mu^2\sqrt{\rho\varepsilon}} && \left(\chi^2 = \sigma^2 + \frac{c_r\varepsilon^2}{L\eta}\right)\\
&\leq \frac{c_{\mathcal{I}}(1-\mu)}{\mu^2\sqrt{\rho\varepsilon}}\left(\eta_\mu^2 L^2\sigma^2 + c_r\eta_\mu L\varepsilon^2\right) && (\eta \leq \eta_\mu)\\
&\leq 2c_{\mathcal{I}}c_\eta\frac{\sqrt{\varepsilon^3}}{\sqrt{\rho}} && \left(\eta_\mu \leq \frac{\mu\varepsilon}{\sqrt{1-\mu}L\sigma} \text{ and } \eta_\mu \leq \frac{\mu^2\sqrt{\varepsilon}}{1-\mu}\right)
\end{aligned}$$

To guarantee that the sum of these terms is at most $\mathcal{F}$, it suffices to select parameters so that $c_\eta + c_r^2 + c_{\mathcal{I}}c_\eta \leq c_{\mathcal{F}}/2$. $\square$

**Corollary B.6** *Let $t_0$ be an iteration of Algorithm 1 such that $\mathbf{e}_{t_0} = 0$. Under Assumptions A and B, for $\mathcal{F}, \mathcal{R}, \mathcal{I}$ chosen as specified in Definition B.2, if there exists $t \in [0, \mathcal{I}]$ such that $\mathbb{E}_{t_0}\left[\|\mathbf{y}_{t_0+t} - \mathbf{y}_{t_0}\|^2\right] > \mathcal{R}^2$, then $f(\mathbf{y}_{t_0}) - \mathbb{E}_{t_0}\left[f(\mathbf{y}_{t_0+t})\right] \geq \mathcal{F}$.*

**Proof :** By Lemma B.4, since $\mathcal{R} = c_\mathcal{R}\sqrt{\frac{\varepsilon}{\rho}}$ and $\mathcal{F} = c_\mathcal{F}\sqrt{\frac{\varepsilon^3}{\rho}}$:

$$f(\mathbf{y}_{t_0}) - \mathbb{E}_{t_0}\left[f(\mathbf{y}_{t_0+t})\right] \geq \frac{\sqrt{\rho\varepsilon}\mathcal{R}^2}{8c_\mathcal{I}} - \mathcal{F} = \frac{c_\mathcal{R}^2\varepsilon\eta\sqrt{\rho\varepsilon}}{8c_\mathcal{I}\eta\rho} - \mathcal{F} = \left(\frac{c_\mathcal{R}^2}{8c_\mathcal{I}c_\mathcal{F}} - 1\right)\mathcal{F} \geq \mathcal{F},$$

where the last inequality holds when $16c_\mathcal{I}c_\mathcal{F} \leq c_\mathcal{R}^2$. $\qquad\square$

## B.3  Large gradient case: $\|\nabla f(\mathbf{x}_{t_0})\| \geq 4L\mathcal{R}$

In this section, we consider the case when the gradient is large, and therefore we can make sufficient progress simply by the Compressed Descent Lemma. Note that the results from this section are only required when the compressor is not linear.

---

**Lemma B.7 (Large gradient case)** *Let $t_0$ be an iteration of Algorithm 1 such that $\mathbf{e}_{t_0} = 0$. Under Assumptions A and B, for $\mathcal{F}, \mathcal{R}, \mathcal{I}$ chosen as specified in Definition B.2, if $\|\nabla f(\mathbf{x}_{t_0})\| > 4L\mathcal{R}$, then after at most $\mathcal{I}$ iterations the objective decreases by $\mathcal{F}$.*

---

**Proof :** Using notation $\mathbb{E}_t\left[\cdot\right]$ from Definition A.4, if there exists $t \leq \mathcal{I}$ such that $\mathbb{E}_{t_0}\left[\|\mathbf{y}_{t_0+t} - \mathbf{y}_{t_0}\|^2\right] > \mathcal{R}^2$, then by Corollary B.6, the objective decreases by at least $\mathcal{F}$.

Consider the case when $\mathbb{E}_{t_0}\left[\|\mathbf{y}_{t_0+t} - \mathbf{y}_{t_0}\|^2\right] \leq \mathcal{R}^2$ for all $t$. First, to bound the error term, we show by induction that $\mathbb{E}_{t_0}\left[\|\nabla f(\mathbf{x}_{t_0+t})\|^2\right] \leq 4\|\nabla f(\mathbf{x}_{t_0})\|^2$ for all $t \leq \mathcal{I}$.

$$
\begin{aligned}
\|\nabla f(\mathbf{x}_{t_0+t})\|^2 &= \|\nabla f(\mathbf{x}_{t_0}) - (\nabla f(\mathbf{x}_{t_0}) - \nabla f(\mathbf{x}_{t_0+t}))\|^2 \\
&\leq 2\|\nabla f(\mathbf{x}_{t_0})\|^2 + 2\|\nabla f(\mathbf{x}_{t_0}) - \nabla f(\mathbf{x}_{t_0+t})\|^2 && (\|a+b\|^2 \leq 2(\|a\|^2 + \|b\|^2)) \\
&\leq 2\|\nabla f(\mathbf{x}_{t_0})\|^2 + 2L^2\|\mathbf{x}_{t_0} - \mathbf{x}_{t_0+t}\|^2 && (\text{Smoothness}) \\
&\leq 2\|\nabla f(\mathbf{x}_{t_0})\|^2 + 4L^2\|\mathbf{y}_{t_0} - \mathbf{y}_{t_0+t}\|^2 + 4L^2\|\mathbf{y}_{t_0+t} - \mathbf{x}_{t_0+t}\|^2 && (\text{Same inequality and } \mathbf{x}_{t_0} = \mathbf{y}_{t_0}) \\
&\leq 2\|\nabla f(\mathbf{x}_{t_0})\|^2 + 4L^2\|\mathbf{y}_{t_0} - \mathbf{y}_{t_0+t}\|^2 + 4L^2\eta^2\|\mathbf{e}_{t_0+t}\|^2 && (\text{Definition A.2 of } \mathbf{y}_{t_0+t})
\end{aligned}
$$

By Lemma A.5 and the induction hypothesis, we have:

$$\mathbb{E}_{t_0}\left[\|\mathbf{e}_{t_0+t}\|^2\right] \leq \frac{4(1-\mu)}{\mu^2}\left(\max_{\tau=0}^{t-1}\mathbb{E}_{t_0}\left[\|\nabla f(\mathbf{x}_{t_0+\tau})\|^2\right] + \chi^2\right) \leq \frac{4(1-\mu)}{\mu^2}\left(4\|\nabla f(\mathbf{x}_{t_0})\|^2 + \chi^2\right),$$

and therefore for $\eta$ chosen as in Definition B.2, $L^2\eta^2\mathbb{E}_{t_0}\left[\|\mathbf{e}_{t_0+t}\|^2\right] \leq \frac{\|\nabla f(\mathbf{x}_{t_0})\|^2}{4}$. By taking the expectation in the equation above, we have:

$$\mathbb{E}_{t_0}\left[\|\nabla f(\mathbf{x}_{t_0+t})\|^2\right] \leq 2\|\nabla f(\mathbf{x}_{t_0})\|^2 + 4L^2\mathcal{R}^2 + \frac{\|\nabla f(\mathbf{x}_{t_0})\|^2}{4} \leq 4\|\nabla f(\mathbf{x}_{t_0})\|^2$$

Given the bound on $\|\mathbf{e}_{t_0+t}\|$, we can give a lower bound on gradient norm:

$$
\begin{aligned}
\|\nabla f(\mathbf{x}_{t_0+t})\|^2 &= \|\nabla f(\mathbf{x}_{t_0}) - (\nabla f(\mathbf{x}_{t_0}) - \nabla f(\mathbf{x}_{t_0+t}))\|^2 \\
&\geq \|\nabla f(\mathbf{x}_{t_0})\|^2 + \|\nabla f(\mathbf{x}_{t_0}) - \nabla f(\mathbf{x}_{t_0+t})\|^2 - 2\|\nabla f(\mathbf{x}_{t_0})\| \cdot \|\nabla f(\mathbf{x}_{t_0}) - \nabla f(\mathbf{x}_{t_0+t})\| \\
&\geq \|\nabla f(\mathbf{x}_{t_0})\|(\|\nabla f(\mathbf{x}_{t_0})\| - 2\|\nabla f(\mathbf{y}_{t_0}) - \nabla f(\mathbf{y}_{t_0+t})\| - 2\|\nabla f(\mathbf{y}_{t_0+t}) - \nabla f(\mathbf{x}_{t_0+t})\|)
\end{aligned}
$$

By taking expectations and using the fact that $\mathbb{E}_{t_0}\left[\|x\|\right] \leq \sqrt{\mathbb{E}_{t_0}\left[\|x\|^2\right]}$ and bound on $\|\mathbf{e}_{t_0+t}\|$, we have:

$$\mathbb{E}_{t_0}\left[\|\nabla f(\mathbf{x}_{t_0+t})\|^2\right] \geq \nabla f(\mathbf{x}_{t_0})(\|\nabla f(\mathbf{x}_{t_0})\| - 2L\mathcal{R} - \frac{\|\nabla f(\mathbf{x}_{t_0})\|}{4}) \geq 4L^2\mathcal{R}^2$$

By Lemma A.7, we know:

$$\sum_{\tau=0}^{\mathcal{I}} \mathbb{E}_{t_0}\left[\|\nabla f(\mathbf{x}_{t_0+\tau})\|^2\right] \leq \frac{4(f(\mathbf{y}_{t_0}) - \mathbb{E}_{t_0}[f(\mathbf{y}_{\mathcal{I}})])}{\eta} + \eta\chi^2\mathcal{I}\left(2L + \frac{8(1-\mu)L^2\eta}{\mu^2}\right)$$

Therefore:

$$\begin{aligned}
f(\mathbf{y}_{t_0}) - \mathbb{E}_{t_0}[f(\mathbf{y}_{t_0+\mathcal{I}})] &\geq \frac{\eta\mathcal{I}}{4}\left(4L^2\mathcal{R}^2 - \eta\chi^2(2L + \frac{8(1-\mu)L^2\eta}{\mu^2})\right) \\
&\geq \eta\mathcal{I}L^2\mathcal{R}^2 - \mathcal{F} && \text{(See proof of Corollary B.5)} \\
&\geq \frac{c_{\mathcal{I}}\eta}{\eta\sqrt{\rho\varepsilon}}\frac{c_{\mathcal{R}}^2 L^2\varepsilon}{\rho} - \mathcal{F} && (\mathcal{R} = c_{\mathcal{R}}\sqrt{\frac{\varepsilon}{\rho}} \text{ and } \mathcal{I} = \frac{c_{\mathcal{I}}}{\eta\sqrt{\rho\varepsilon}}) \\
&\geq \frac{c_{\mathcal{I}}}{\sqrt{\rho\varepsilon}}c_{\mathcal{R}}^2\varepsilon^2 - \mathcal{F} && \text{(since } L \geq \sqrt{\rho\varepsilon}) \\
&\geq \mathcal{F},
\end{aligned}$$

where the last inequality holds when $c_{\mathcal{I}}c_{\mathcal{R}}^2 \geq 2c_{\mathcal{F}}$. $\qquad\square$

## B.4    Small Gradient Case: $\|\nabla f(\mathbf{x}_{t_0})\| < 4L\mathcal{R}$

### Coupling Sequences

Let $H = \nabla^2 f(\mathbf{x}_{t_0})$, then $g(\mathbf{x}) = \mathbf{x}^\top H\mathbf{x}$ is a quadratic approximation of $f$ in the vicinity of $\mathbf{x}_{t_0}$. Let $-\gamma$ be the smallest eigenvalue of $H$ and $\mathbf{v}_1$ be the corresponding eigenvector. Then we construct *coupling sequences* $\mathbf{x}_{t_0+t}$ and $\mathbf{x}'_{t_0+t}$ in the following way: $\mathbf{x}_{t_0+t}$ is the sequence from Algorithm 1; $\mathbf{x}'_{t_0+t}$ has the same stochastic randomness $\theta$ as $\mathbf{x}_{t_0+t}$, and its artificial noise $\xi'_{t_0+t}$ is the same as $\xi_{t_0+t}$ with exception of the coordinate corresponding to $\mathbf{v}_1$, which has an opposite sign.

---

**Definition B.8 (Coupling sequences)** *For iteration $t_0$, given $\mathbf{x}_{t_0}$ and $\mathbf{e}_{t_0}$, the coupling sequences are defined as follows (note the definition of $\xi'_{t_0+t}$):*

$$\mathbf{e}'_{t_0} = \mathbf{e}_{t_0}$$

$$\begin{aligned}
\xi_{t_0+t} &\sim \mathcal{N}(0, r^2) & \xi'_{t_0+t} &= \xi_{t_0+t} - 2\langle\mathbf{v}_1, \xi_{t_0+t}\rangle\mathbf{v}_1 \\
\theta_{t_0+t} &\sim \mathcal{D}, \quad \tilde{\theta}_{t_0+t} \sim \tilde{\mathcal{D}} & \theta'_{t_0+t} &= \theta_{t_0+t}, \quad \tilde{\theta}'_{t_0+t} = \tilde{\theta}_{t_0+t} \\
\mathbf{g}_{t_0+t} &= C(\nabla F(\mathbf{x}_{t_0+t}, \theta_{t_0+t}) + \xi_{t_0+t} + \mathbf{e}_{t_0+t}, \tilde{\theta}_{t_0+t}) & \mathbf{g}'_{t_0+t} &= C(\nabla F(\mathbf{x}'_{t_0+t}, \theta_{t_0+t}) + \xi'_{t_0+t} + \mathbf{e}'_{t_0+t}, \tilde{\theta}_{t_0+t}) \\
\mathbf{y}_{t_0+t} &= \mathbf{x}_{t_0+t} - \eta\mathbf{e}_{t_0+t} & \mathbf{y}'_{t_0+t} &= \mathbf{x}'_{t_0+t} - \eta\mathbf{e}'_{t_0+t} \\
\mathbf{x}_{t+1} &= \mathbf{x}_{t_0+t} - \eta\mathbf{g}_t & \mathbf{x}'_{t+1} &= \mathbf{x}'_{t_0+t} - \eta\mathbf{g}'_t \\
\mathbf{e}_{t+1} &= \nabla F(\mathbf{x}{t_0} + t, \theta_{t_0+t}) + \xi_{t_0+t} + \mathbf{e}_{t_0+t} - \mathbf{g}_{t_0+t} & \mathbf{e}'_{t+1} &= \nabla F(x'_t, \theta_{t_0+t}) + \xi'_{t_0+t} + \mathbf{e}'_{t_0+t} - \mathbf{g}'_{t_0+t}
\end{aligned}$$

---

A notable fact is that both sequences correspond to the same distribution.

**Proposition B.9** *For any $t_0$ and $t$, $\mathbf{x}_{t_0+t}$ and $\mathbf{y}_{t_0+t}$ from Definition B.8 have the same distribution as $\mathbf{x}'_{t_0+t}$ and $\mathbf{y}'_{t_0+t}$.*

**Proof :**     By definition of $\mathbf{y}_{t_0+t}$ and $\mathbf{y}'_{t_0+t}$, it suffices show that $\mathbf{x}_{t_0+t}$ and $\mathbf{e}_{t_0+t}$ have the same distributions as $\mathbf{x}'_{t_0+t}$ and $\mathbf{e}'_{t_0+t}$. Proof by Induction with trivial base case $\mathbf{y}_{t_0} = \mathbf{y}'_{t_0} = \mathbf{x}_{t_0} - \eta\mathbf{e}_{t_0}$.

We want to show that if the statement holds for $t$, then it holds for $t+1$. To show that $\mathbf{x}_{t_0+t+1}$ and $\mathbf{x}'_{t_0+t+1}$ habe the same distribution it remains to show that $\mathbf{g}_t$ and $\mathbf{g}'_t$ have the same distribution:

- Since $\mathbf{x}_{t_0+t}$ and $\mathbf{x}'_{t_0+t}$ have the same distribution, $\nabla F(\mathbf{x}_{t_0+t}, \theta_{t_0+t})$ and $\nabla F(\mathbf{x}'_{t_0+t}, \theta_{t_0+t})$ have the same distribution.
- Since $\mathcal{N}(0, r^2)$ is symmetric and $\xi'_{t_0+t}$ is the same as $\xi_{t_0+t}$ with exception of one coordinate, which has an opposite sign, $\xi_{t_0+t}$ and $\xi'_{t_0+t}$ have the same distribution.
- $\mathbf{e}_{t_0+t}$ and $\mathbf{e}'_{t_0+t}$ have the same distribution.

Similarly, $\mathbf{e}_{t+1}$ has the same distribution as $\mathbf{e}'_{t+1}$, since $\nabla F(\mathbf{x}_{t_0+t}, \theta_{t_0+t})$, $\xi_{t_0+t}$, $\mathbf{e}_{t_0+t}$ and $\mathbf{g}_t$ have the same distribution as $\nabla F(\mathbf{x}'_{t_0+t}, \theta_{t_0+t})$, $\xi'_{t_0+t}$, $\mathbf{e}'_{t_0+t}$ and $\mathbf{g}'_t$. $\qquad\square$

Since our sequences have the same distribution, we have $\mathbb{E}_{t_0}\left[f(\mathbf{x}_{t_0+t})\right] = \mathbb{E}_{t_0}\left[f(\mathbf{x}'_{t_0+t})\right]$. We want to show that in a few iterations $\mathbf{y}'_{t_0+t} - \mathbf{y}_{t_0+t}$ becomes sufficiently large and, therefore, at least one of $\mathbf{y}_{t_0+t}$ and $\mathbf{y}'_{t_0+t}$ is far from $\mathbf{x}_{t_0}$. By applying Corollary B.6 we will show that the objective sufficiently decreases.

## Difference Between Coupling Sequences

In order to capture the difference between the coupling sequences, we introduce the following notation:

---

**Definition B.10 (Difference between sequences)** *Using notation from Definition B.8, we introduce differences between the sequences:*

$$\hat{\mathbf{x}}_{t_0+t} = \mathbf{x}'_{t_0+t} - \mathbf{x}_{t_0+t} \qquad \hat{\mathbf{e}}_{t_0+t} = \mathbf{e}'_{t_0+t} - \mathbf{e}_{t_0+t} \qquad \hat{\zeta}_t = \zeta'_{t_0+t} - \zeta_{t_0+t}$$

$$\hat{\xi}_{t_0+t} = \xi'_{t_0+t} - \xi_{t_0+t} \qquad \hat{\mathbf{y}}_{t_0+t} = \mathbf{y}'_{t_0+t} - \mathbf{y}_{t_0+t}$$

---

**Definition B.11 (Error terms)** *Let $\delta_i = \int_0^1 \nabla^2 f(\alpha \mathbf{x}'_{t_0+i} + (1-\alpha)\mathbf{x}_{t_0+i})d\alpha - H$. Then*

$$\Delta_t = \eta \sum_{i=0}^{t-1} (I - \eta H)^{t-i-1} \delta_i \hat{\mathbf{x}}_{t_0+i}$$

$$\mathcal{E}_t = \eta \sum_{i=0}^{t-1} (I - \eta H)^{t-i-1} (\hat{\mathbf{e}}_{t_0+i} - \hat{\mathbf{e}}_{t_0+i+1})$$

$$Z_t = \eta \sum_{i=0}^{t-1} (I - \eta H)^{t-i-1} \hat{\zeta}_{t_0+i}$$

$$\Xi_t = \eta \sum_{i=0}^{t-1} (I - \eta H)^{t-i-1} \hat{\xi}_{t_0+i},$$

---

**Proposition B.12** *For any $t$: $\hat{\mathbf{x}}_{t_0+t} = -(\Delta_t + \mathcal{E}_t + Z_t + \Xi_t)$.*

In the simplest case, the objective is quadratic and we have access to an uncompressed deterministic gradient. When it's not the case, the introduced terms show how the actual algorithm behavior is different:

- $\Delta_t$ corresponds to quadratic approximation error.
- $\mathcal{E}_t$ corresponds to compression error.
- $Z_t$ corresponds to difference arising from SGD noise.
- $\Xi_t$ corresponds to difference arising from artificial noise.

Intuitively, $\Xi_t$ is a good term, and other terms are negligible ($\|\Delta_t + \mathcal{E}_t + Z_t\| < \frac{1}{2}\|\Xi_t\|$).

**Proof :**

$$
\begin{aligned}
&\hat{\mathbf{x}}_{t_0+t+1}\\
&= \mathbf{x}'_{t_0+t+1} - \mathbf{x}_{t_0+t+1}\\
&= \mathbf{y}'_{t_0+t+1} + \eta\mathbf{e}'_{t_0+t+1} - (\mathbf{y}_{t_0+t+1} + \eta\mathbf{e}_{t_0+t+1}) && \text{(Def. of } \mathbf{y}_{t_0+t} \text{ and } \mathbf{y}'_{t_0+t})\\
&= \eta\hat{\mathbf{e}}_{t_0+t+1} + (\mathbf{y}'_{t_0+t} - \mathbf{y}_{t_0+t})\\
&\quad - \eta\left((\nabla f(x'_{t_0+t}) - \nabla f(x_{t_0+t})) + (\zeta'_{t_0+t} - \zeta_{t_0+t}) + (\xi'_{t_0+t} - \xi_{t_0+t})\right) && \text{(Upd. equation for } \mathbf{y}_{t_0+t})\\
&= \eta(\hat{\mathbf{e}}_{t+1} - \hat{\mathbf{e}}_{t_0+t}) + \hat{\mathbf{x}}_{t_0+t} - \eta\left((\delta_t + H)\hat{\mathbf{x}}_{t_0+t} + \hat{\zeta}_{t_0+t} + \hat{\xi}_{t_0+t}\right) && \text{(Def. of } \delta_{t_0+t} \text{ and } \mathbf{y}_{t_0+t})\\
&= \eta(\hat{\mathbf{e}}_{t+1} - \hat{\mathbf{e}}_{t_0+t}) + (I - \eta H)\hat{\mathbf{x}}_{t_0+t} - \eta\left(\delta_t\hat{\mathbf{x}}_{t_0+t} + \hat{\zeta}_{t_0+t} + \hat{\xi}_{t_0+t}\right)\\
&= (I - \eta H)\hat{\mathbf{x}}_{t_0+t} - \eta\left(\delta_t\hat{\mathbf{x}}_{t_0+t} + (\hat{\mathbf{e}}_{t_0+t} - \hat{\mathbf{e}}_{t_0+t+1}) + \hat{\zeta}_{t_0+t} + \hat{\xi}_{t_0+t}\right)
\end{aligned}
$$

Using telescoping, we get the required expression. $\qquad\square$

Since $\hat{\mathbf{y}}_{t_0+t} = \hat{\mathbf{x}}_{t_0+t} - \eta\hat{\mathbf{e}}_{t_0+t}$, we have:

$$
\hat{\mathbf{x}}_{t_0+t} = -(\Delta_t + \mathcal{E}_t + Z_t + \Xi_t) \iff \hat{\mathbf{y}}_{t_0+t} = -(\Delta_t + (\mathcal{E}_t + \eta\hat{\mathbf{e}}_{t_0+t}) + Z_t + \Xi_t),
$$

and we'll use $\|\hat{\mathbf{y}}_{t_0+t}\|$ in Corollary B.6.

### Bounding Accumulated Compression Error

Compared to SGD analysis, an additional term $\mathcal{E}_t + \eta\hat{\mathbf{e}}_{t_0+t}$ appears. This term corresponds to accumulated error arising from compression, and we have to bound it. Motivated by Jin et al. [2021], we introduce the following quantity:

---

**Definition B.13** *Standard deviation of sum of random variables with standard deviations* $(1+\eta\gamma)^i$, $i = 0, \ldots, t-1$, *is*

$$
\beta_t = \sqrt{\sum_{i=0}^{t-1}(1+\eta\gamma)^{2i}}
$$

---

**Proposition B.14 (Jin et al. [2021], Lemma 29)** *If* $\eta\gamma \in [0,1]$, *then for all* $t$: $\beta_t \le \frac{(1+\eta\gamma)^t}{\sqrt{2\eta\gamma}}$, *and for all* $t \ge \frac{2}{\eta\gamma}$: $\beta_t \ge \frac{(1+\eta\gamma)^t}{\sqrt{6\eta\gamma}}$.

**Proposition B.15** *For any* $t \le \mathcal{I}$, *where* $\mathcal{I}$ *is defined in Definition B.2:*

$$
\left(\sum_{i=0}^{t-1}(1+\eta\gamma)^{t-1-i}\right)^2 \le \frac{c_{\mathcal{I}}\beta_t^2}{\eta\sqrt{\rho\varepsilon}}
$$

**Proof :** By Cauchy-Schwarz inequality:

$$
\left(\sum_{i=0}^{t-1}(1+\eta\gamma)^{t-1-i}\right)^2 \le t\sum_{i=0}^{t-1}(1+\eta\gamma)^{2(t-1-i)} \le \mathcal{I}\beta_t^2 = \frac{c_{\mathcal{I}}\beta_t^2}{\eta\sqrt{\rho\varepsilon}}
$$

$\qquad\square$

**Lemma B.16 (Bounding accumulated compression error)** *Let $t_0$ be an iteration such that $\mathbf{e}_{t_0} = 0$ in Algorithm 1. Under Assumptions A and B, let $\chi$ be as in Definition A.1, $\eta$ and $\mathcal{R}$ as in Definition B.2, $\mathcal{E}_t$ and $\hat{\mathbf{e}}_{t_0+t}$ be as in Definition B.11, and $\beta_t$ be as in Definition B.13 . Let $-\gamma \leq -\sqrt{\rho\varepsilon}/2$ be the smallest negative eigenvalue of $\nabla^2 f(\mathbf{x}_{t_0})$. If $\mathbb{E}_{t_0}\left[\|\mathbf{y}_{t_0+t} - \mathbf{y}_{t_0}\|^2\right] < \mathcal{R}^2$ for all $t \leq \mathcal{I}$ and $\|\nabla f(\mathbf{x}_{t_0})\| \leq 4L\mathcal{R}$, then using notation $\mathbb{E}_t[\cdot]$ from Definition A.4, for $t \leq \mathcal{I}$ we have:*

$$\mathbb{E}_{t_0}\left[\|\mathcal{E}_t + \eta\hat{\mathbf{e}}_{t_0+t}\|^2\right] \leq \frac{20c_{\mathcal{I}}\eta^3(1-\mu)L^2\chi^2\beta_t^2}{\mu^2\sqrt{\rho\varepsilon}}$$

**Proof :**  Expanding sum in $\mathcal{E}_t$ and using that $\hat{\mathbf{e}}_{t_0} = 0$:

$$\mathcal{E}_t = \eta\sum_{i=0}^{t-1}(I - \eta H)^{t-1-i}(\hat{\mathbf{e}}_{t_0+i} - \hat{\mathbf{e}}_{t_0+i+1}) \qquad \text{(By Definition B.11)}$$

$$= \eta(-\hat{\mathbf{e}}_{t_0+t} + \sum_{i=1}^{t-1}(I - \eta H)^{t-1-i}((I - \eta H) - I)\hat{\mathbf{e}}_{t_0+i}) \qquad \text{(By telescoping)}$$

$$= -\eta\hat{\mathbf{e}}_{t_0+t} + \eta^2 H\sum_{i=1}^{t-1}(I - \eta H)^{t-1-i}\hat{\mathbf{e}}_{t_0+i}$$

We will now estimate $\mathbb{E}_{t_0}\left[\|\mathcal{E}_t + \eta\hat{\mathbf{e}}_{t_0+t}^2\|\right]$. Since $-\gamma$ is the smallest negative eigenvalue of $H$, we have $\|I - \eta H\| \leq (1 + \eta\gamma)$.

$$\mathbb{E}_{t_0}\left[\|\mathcal{E}_t + \eta\hat{\mathbf{e}}_{t_0+t}\|^2\right]$$

$$= \mathbb{E}_{t_0}\left[\|\eta^2 H\sum_{i=1}^{t-1}(I - \eta H)^{t-1-i}\hat{\mathbf{e}}_{t_0+i}\|^2\right]$$

$$\leq \eta^4 L^2\mathbb{E}_{t_0}\left[(\sum_i(1 + \eta\gamma)^{t-1-i}\|\hat{\mathbf{e}}_{t_0+i}\|)^2\right] \qquad \text{(By } L\text{-smoothness, } \lambda_{\max}(H) \leq L)$$

$$\leq 2\eta^4 L^2(\sum_i(1 + \eta\gamma)^{t-1-i})^2\max_i\mathbb{E}_{t_0}\left[\|\hat{\mathbf{e}}_{t_0+i}\|^2\right] \qquad (\mathbb{E}_{t_0}[ab] \leq \max(\mathbb{E}_{t_0}\left[a^2\right], \mathbb{E}_{t_0}\left[b^2\right]))$$

$$\leq 2\eta^4 L^2 t(\sum_i(1 + \eta\gamma)^{t-1-i})^2\max_i\mathbb{E}_{t_0}\left[\|\mathbf{e}'_{t_0+i} - \mathbf{e}_{t_0+i}\|^2\right] \qquad \text{(By definition of } \hat{\mathbf{e}}_{t_0+i})$$

$$\leq 4\eta^4 L^2 t(\sum_i(1 + \eta\gamma)^{t-1-i})^2\max_i\mathbb{E}_{t_0}\left[\|\mathbf{e}'_{t_0+i}\|^2 + \|\mathbf{e}_{t_0+i}\|^2\right] \qquad \text{(By Cauchy-Schwarz)}$$

$$\leq 8\eta^4 L^2 t(\sum_i(1 + \eta\gamma)^{t-1-i})^2\max_i\mathbb{E}_{t_0}\left[\|\mathbf{e}_{t_0+i}\|^2\right] \qquad (\mathbf{e}_{t_0+i} \text{ and } \mathbf{e}'_{t_0+i} \text{ have the same distribution})$$

Similarly to Lemma B.7, we can show that $\mathbb{E}_{t_0}\left[\|\nabla f(\mathbf{x}_{t_0+i})\|^2\right] \leq 40L^2\mathcal{R}^2$. By Lemma A.5:

$$\mathbb{E}_{t_0}\left[\|\mathbf{e}_{t_0+t}\|^2\right]$$

$$\leq \frac{4(1-\mu)}{\mu^2}(\max_i \mathbb{E}_{t_0}\left[\|\nabla f(\mathbf{x}_{t_0+i})\|^2\right] + \chi^2) \qquad\qquad \text{(By Lemma A.5)}$$

$$\leq \frac{4(1-\mu)}{\mu^2}(40L^2\mathcal{R}^2 + \chi^2) \qquad\qquad \text{(By assumption } \mathbb{E}_{t_0}\left[\|\nabla f(\mathbf{x}_{t_0+i})\|^2\right] \leq 40L^2\mathcal{R}^2\text{)}$$

$$\leq \frac{5(1-\mu)\chi^2}{\mu^2} \qquad\qquad \text{(Selecting sufficiently small } c_\mathcal{R} \text{ in the definition of } \mathcal{R}\text{)}$$

Substituting this result into the inequality for $\|\mathcal{E}_t + \eta\hat{\mathbf{e}}_{t_0+t}\|$:

$$\mathbb{E}_{t_0}\left[\|\mathcal{E}_t + \eta\hat{\mathbf{e}}_{t_0+t}\|^2\right] \leq 4\eta^4 L^2 t \left(\sum_i (1+\eta\gamma)^{t-1-i}\right)^2 \frac{5(1-\mu)\chi^2}{\mu^2} \leq \frac{20c_\mathcal{I}\eta^4(1-\mu)L^2\chi^2\beta_t^2}{\mu^2\eta\sqrt{\rho\varepsilon}},$$

where we bounded the series using Proposition B.15. $\qquad\qquad\square$

---

**Lemma B.17 (Bounding accumulated compression error for linear compressor)** *Under conditions of Lemma B.16, additionally assume that the compressor is linear (Definition 2.2). When $\eta \leq \eta_\sigma$, for $t \leq \mathcal{I}$:*

$$\mathbb{E}_{t_0}\left[\|\mathcal{E}_t + \eta\hat{\mathbf{e}}_{t_0+t}\|^2\right] \leq \frac{9c_\mathcal{I}\eta^3(1-\mu)L^2\beta_t^2 r^2}{\mu^2 d\sqrt{\rho\varepsilon}}$$

---

Note that, compared with Lemma B.16, the bound is improved by the factor of $d$.

**Proof :**

$$\hat{\mathbf{e}}_{t_0+t+1} = \mathbf{e}_{t_0+t+1} - \mathbf{e}'_{t_0+t+1}$$

$$= \nabla F(\mathbf{x}_{t_0+t}, \theta_{t_0+t}) + \xi_{t_0+t} + \mathbf{e}_{t_0+t} - \mathcal{C}(\nabla F(\mathbf{x}_{t_0+t}, \theta_{t_0+t}) + \xi_{t_0+t} + \mathbf{e}_{t_0+t}, \tilde{\theta}_{t_0+t})$$

$$\quad - (\nabla F(\mathbf{x}'_{t_0+t}, \theta_{t_0+t}) + \xi'_{t_0+t} + \mathbf{e}'_{t_0+t} - \mathcal{C}(\nabla F(\mathbf{x}'_{t_0+t}, \theta_{t_0+t}) + \xi'_{t_0+t} + \mathbf{e}'_{t_0+t}, \tilde{\theta}_{t_0+t}))$$

$$= (\nabla F(\mathbf{x}_{t_0+t}, \theta_{t_0+t}) - \nabla F(\mathbf{x}'_{t_0+t}, \theta_{t_0+t})) + (\xi_{t_0+t} - \xi'_{t_0+t}) + (\mathbf{e}_{t_0+t} - \mathbf{e}'_{t_0+t})$$

$$\quad - \mathcal{C}\left((\nabla F(\mathbf{x}_{t_0+t}, \theta_{t_0+t}) - \nabla F(\mathbf{x}'_{t_0+t}, \theta_{t_0+t})) + (\xi_{t_0+t} - \xi'_{t_0+t}) + (\mathbf{e}_{t_0+t} - \mathbf{e}'_{t_0+t}), \tilde{\theta}_{t_0+t}\right)$$

$$= (\nabla F(\mathbf{x}_{t_0+t}, \theta_{t_0+t}) - \nabla F(\mathbf{x}'_{t_0+t}, \theta_{t_0+t})) + \hat{\xi}_{t_0+t} + \hat{\mathbf{e}}_{t_0+t}$$

$$\quad - \mathcal{C}\left((\nabla F(\mathbf{x}_{t_0+t}, \theta_{t_0+t}) - \nabla F(\mathbf{x}'_{t_0+t}, \theta_{t_0+t})) + \hat{\xi}_{t_0+t} + \hat{\mathbf{e}}_{t_0+t}, \tilde{\theta}_{t_0+t}\right)$$

We estimating the norm of $\hat{\mathbf{e}}_{t_0+t}$ using linearity of $\mathcal{C}$:

$$\mathbb{E}_{\tilde{\theta}_{t_0+t}}\left[\|\hat{\mathbf{e}}_{t+1}\|^2 \mid \mathbf{x}_{t_0+t}, \mathbf{e}_{t_0+t}, \theta_{t_0+t}\right]$$

$$= \mathbb{E}_{\tilde{\theta}_{t_0+t}}\Big[\|(\nabla F(\mathbf{x}_{t_0+t}, \theta_{t_0+t}) - \nabla F(\mathbf{x}'_{t_0+t}, \theta_{t_0+t})) + \hat{\xi}_{t_0+t} + \hat{\mathbf{e}}_{t_0+t}$$

$$\quad - \mathcal{C}\left((\nabla F(\mathbf{x}_{t_0+t}, \theta_{t_0+t}) - \nabla F(\mathbf{x}'_{t_0+t}, \theta_{t_0+t})) + \hat{\xi}_{t_0+t} + \hat{\mathbf{e}}_{t_0+t}, \tilde{\theta}_{t_0+t}\right)\|^2\Big]$$

$$\leq (1-\mu)\mathbb{E}_{\tilde{\theta}_{t_0+t}}\left[\|(\nabla F(\mathbf{x}_{t_0+t}, \theta_{t_0+t}) - \nabla F(\mathbf{x}'_{t_0+t}, \theta_{t_0+t})) + \hat{\xi}_{t_0+t} + \hat{\mathbf{e}}_{t_0+t}\|^2\right]$$

Similarly to the proof of Lemma A.5, for any $\nu$ we have:

$$\mathbb{E}_{t_0}\left[\|\hat{\mathbf{e}}_{t_0+t+1}\|^2\right]$$

$$\leq (1-\mu)\mathbb{E}_{t_0}\left[(1+\frac{1}{\nu})\|(\nabla F(\mathbf{x}_{t_0+t},\theta_{t_0+t}) - \nabla F(\mathbf{x}'_{t_0+t},\theta_{t_0+t})) + \hat{\xi}_{t_0+t}\|^2 + (1+\nu)\|\hat{\mathbf{e}}_{t_0+t}\|^2\right]$$

$$\leq \frac{1}{\nu}\sum_{i=0}^{t}((1-\mu)(1+\nu))^{t-i+1}\,\mathbb{E}_{t_0}\left[\|(\nabla F(\mathbf{x}_{t_0+t},\theta_{t_0+t}) - \nabla F(\mathbf{x}'_{t_0+t},\theta_{t_0+t})) + \hat{\xi}_{t_0+t}\|^2\right]$$

By selecting $\nu = \frac{\mu}{2(1-\mu)}$ and computing the sum of a geometric series, we have:

$$\mathbb{E}_{t_0}\left[\|\hat{\mathbf{e}}_{t_0+t+1}\|^2\right] \leq \frac{2(1-\mu)}{\mu}\sum_{i=0}^{t}\left(1-\frac{\mu}{2}\right)^{t-i+1}\mathbb{E}_{t_0}\left[\|(\nabla F(\mathbf{x}_{t_0+t},\theta_{t_0+t}) - \nabla F(\mathbf{x}'_{t_0+t},\theta_{t_0+t})) + \hat{\xi}_{t_0+t}\|^2\right]$$

$$\leq \frac{2(1-\mu)}{\mu}\sum_{i=0}^{t}\left(1-\frac{\mu}{2}\right)^{t-i+1}\max_{i}\mathbb{E}_{t_0}\left[\|(\nabla F(\mathbf{x}_{t_0+i},\theta_{t_0+i}) - \nabla F(\mathbf{x}'_{t_0+i},\theta_{t_0+i})) + \hat{\xi}_{t_0+i}\|^2\right]$$

$$\leq \frac{4(1-\mu)}{\mu^2}\max_{i=0}^{t}\mathbb{E}_{t_0}\left[\|(\nabla F(\mathbf{x}_{t_0+i},\theta_{t_0+i}) - \nabla F(\mathbf{x}'_{t_0+i},\theta_{t_0+i})) + \hat{\xi}_{t_0+i}\|^2\right]$$

$$\leq \frac{4(1-\mu)}{\mu^2}\max_{i=0}^{t}\mathbb{E}_{t_0}\left[\|\nabla F(\mathbf{x}_{t_0+i},\theta_{t_0+i}) - \nabla F(\mathbf{x}'_{t_0+i},\theta_{t_0+i})\|^2\right] + \frac{r^2}{d} + \|\hat{\mathbf{e}}_{t_0}\|^2$$

Substituting this into bound for $\mathcal{E} + \eta\hat{\mathbf{e}}_{t_0+t}$ and bounding the series by Proposition B.15:

$$\mathbb{E}_{t_0}\left[\|\mathcal{E}_t + \eta\hat{\mathbf{e}}_{t_0+t}\|^2\right] \leq 2\eta^4 L^2\left(\sum_{i}(1+\eta\gamma)^{t-1-i}\right)^2 \max_{i=0}^{t-1}\mathbb{E}_{t_0}\left[\|\hat{\mathbf{e}}_{t_0+i}\|^2\right]$$

$$\leq \frac{8c_{\mathcal{I}}\eta^4(1-\mu)L^2\beta_t^2}{\mu^2\eta\sqrt{\rho\varepsilon}}\left(\max_{i=0}^{t-1}\mathbb{E}_{t_0}\left[\|\nabla F(\mathbf{x}_{t_0+i},\theta_{t_0+i}) - \nabla F(\mathbf{x}'_{t_0+i},\theta_{t_0+i})\|^2\right] + \frac{r^2}{d}\right)$$

Depending on whether Assumption C holds, we consider the following cases:

**When Assumption C holds,** we bound $\mathbb{E}_{t_0}\left[\|\nabla F(\mathbf{x}_{t_0+i},\theta_{t_0+i}) - \nabla F(\mathbf{x}'_{t_0+i},\theta_{t_0+i})\|^2\right]$ with $\tilde{\ell}^2\mathbb{E}_{t_0}\left[\|\hat{\mathbf{x}}\|^2\right] \leq 4\tilde{\ell}^2\mathcal{R}^2$. Since $\mathcal{R} = c_{\mathcal{R}}\sqrt{\frac{\varepsilon}{\rho}}$ and $r = c_r\frac{\varepsilon}{\sqrt{L\eta}}$, we can select constants $c_{\mathcal{R}}, c_r$ and $c_\eta$ so that the second term dominates the first one.

**When Assumption C doesn't hold hold,** we use the following bound:

$$\mathbb{E}_{t_0}\left[\|\nabla F(\mathbf{x}_{t_0+i},\theta_{t_0+i}) - \nabla F(\mathbf{x}'_{t_0+i},\theta_{t_0+i})\|^2\right]$$
$$\leq 3\mathbb{E}_{t_0}\left[\|\nabla F(\mathbf{x}_{t_0+i},\theta_{t_0+i}) - \nabla f(\mathbf{x}_{t_0+i})\|^2 + \|\nabla f(\mathbf{x}_{t_0+i}) - \nabla f(\mathbf{x}'_{t_0+i})\|^2 + \|\nabla f(\mathbf{x}'_{t_0+i}) - \nabla F(\mathbf{x}'_{t_0+i},\theta_{t_0+i})\|^2\right]$$
$$\leq 6(\sigma^2 + L^2\mathcal{R}^2)$$

as $c(\sigma^2 + \mathcal{R}^2)$. Using that $\eta \leq \eta_\sigma \leq \frac{\varepsilon^2}{d}$, we again can select the constants so that the second term dominates.

As a result, we achieve the required bound:

$$\mathbb{E}_{t_0}\left[\|\mathcal{E}_t + \eta\hat{\mathbf{e}}_{t_0+t}\|^2\right] \leq \frac{9c_{\mathcal{I}}\eta^3(1-\mu)L^2\beta_t^2 r^2}{\mu^2 d\sqrt{\rho\varepsilon}}$$

$\square$

**Escaping From a Saddle Point**

We now show that, if the starting point is a saddle point, we move sufficiently far from it.

---

**Lemma B.18 (Non-localization)** *Let $t_0$ be an iteration such that $\mathbf{e}_{t_0} = 0$ in Algorithm 1. Under Assumptions A and B, let $\eta$ and $r$ be as in Definition B.2 and $\beta_t$ be as in Definition B.13. Let $\gamma = -\lambda_{\min}(\nabla^2 f(\mathbf{x}_{t_0})) > \frac{\sqrt{\rho\varepsilon}}{2}$ and $\mathbb{E}_{t_0}\left[\|\mathbf{y}'_{t_0+t} - \mathbf{y}_{t_0}\|^2\right] < \mathcal{R}^2$ for all $t \leq \mathcal{I}$. Then for all $t \leq \mathcal{I}$, for some constant $c$:*

$$\mathbb{E}_{t_0}\left[\|\hat{\mathbf{y}}_{t_0+t}\|^2\right] \geq c\frac{\beta_t^2\eta^2 r^2}{d},$$

*where $\hat{\mathbf{y}}_{t_0+t} = \mathbf{y}'_{t_0+t} - \mathbf{y}_{t_0+t}$ as in Definition B.10.*

---

**Proof :** To simplify the presentation, we use $c$ to denote constants, and it may change its meaning from line to line.

$$\mathbb{E}_{t_0}\left[\|\hat{\mathbf{y}}_{t_0+t}\|^2\right] \geq \left(\max\left(0, \mathbb{E}_{t_0}\left[\|\Xi_t\| - \|\Delta_t\| - \|\mathcal{E}_t + \eta\hat{\mathbf{e}}_{t_0+t}\| - \|Z_t\|\right]\right)\right)^2$$

We show that $\mathbb{E}_{t_0}[\Xi_t] = \Omega\left(\frac{\beta_t\eta r}{\sqrt{d}}\right)$, and terms aside from $\Xi_t$ are negligible, namely that in expectation $\mathbb{E}_{t_0}[\|\Delta_t\|], \mathbb{E}_{t_0}[\|\mathcal{E}_t + \eta\hat{\mathbf{e}}_{t_0+t}\|], \mathbb{E}_{t_0}[\|Z_t\|] \leq \frac{1}{10}\mathbb{E}_{t_0}[\|\Xi_t\|]$.[5] We prove the inequality by induction. The inequality holds for $t = 0$ since all terms are 0.

**Estimating $\Xi_t$.** Since $\Xi_t$ is a sum of independent Gaussians with variances $4(1+\eta\gamma)^{2(t-i-1)}\frac{\eta^2 r^2}{d}$, its total variance is

$$\mathbb{E}_{t_0}\left[\|\Xi_t\|^2\right] = 4\frac{\eta^2 r^2}{d}\sum_{i=0}^{t-1}(1+\eta\gamma)^{2i} = 4\frac{\eta^2 r^2}{d}\beta_t^2,$$

And since $\Xi_t$ is a zero-mean Guassian random variable, we know $\mathbb{E}_{t_0}[\|\Xi_t\|]^2 = \frac{2}{\pi}\mathbb{E}_{t_0}[\|\Xi_t\|^2]$. Note that from the induction hypothesis it follows that $\mathbb{E}_{t_0}\left[\|\hat{\mathbf{y}}_{t_0+t}\|^2\right] \leq 2\mathbb{E}_{t_0}\left[\|\Xi_t\|^2\right] \leq 8\frac{\eta^2 r^2\beta_t^2}{d}$.

**Bounding $\Delta_i$.** By the Hessian Lipschitz property, $\mathbb{E}_{t_0}\left[\|\delta_i\|^2\right] \leq 4\rho^2\mathcal{R}^2$. Then for $i \leq t$ and for $\eta$ selected as in Definition B.2 (see proofs of Lemmas B.16 and B.17), by the induction hypothesis:

$$\mathbb{E}_{t_0}\left[\|\hat{\mathbf{x}}_{t_0+i}\|^2\right] \leq 2\mathbb{E}_{t_0}\left[\|\hat{\mathbf{y}}_{t_0+i}\|^2\right] + 2\eta^2\mathbb{E}_{t_0}\left[\|\hat{\mathbf{e}}_{t_0+i}\|^2\right] \leq c\frac{\eta^2 r^2\beta_i^2}{d}$$

---

[5]Most of the proof can go through if we consider $\mathbb{E}_{t_0}\left[\|\cdot\|^2\right]$ instead of $\mathbb{E}_{t_0}[\|\cdot\|]$. There is only one place in the estimation of $\|\Delta_t\|$ which requires the first momentum.

Therefore:

$$\mathbb{E}_{t_0}\left[\|\Delta_t\|\right]$$

$$= \mathbb{E}_{t_0}\left[\|\eta\sum_{i=0}^{t-1}(I-\eta H)^{t-i-1}\delta_i\hat{\mathbf{x}}_{t_0+i}\|\right] \qquad \text{(Definition B.11)}$$

$$\leq \eta\mathbb{E}_{t_0}\left[\sum_{i=0}^{t-1}\|I-\eta H\|^{t-i-1}\cdot\|\delta_i\|\cdot\|\hat{\mathbf{x}}_{t_0+i}\|\right]$$

$$\leq \eta\sum_{i=0}^{t-1}(1+\eta\gamma)^{t-i-1}\sqrt{\mathbb{E}_{t_0}\left[\|\delta_i\|^2\right]\cdot\mathbb{E}_{t_0}\left[\|\hat{\mathbf{x}}_{t_0+i}\|^2\right]} \qquad \text{(Cauchy-Schwarz)}$$

$$\leq c\eta\rho\mathcal{R}\frac{\eta r}{\sqrt{d}}\mathbb{E}_{t_0}\left[\sum_{i=0}^{t-1}(1+\eta\gamma)^{t-i-1}\beta_i\right] \qquad \left(\mathbb{E}_{t_0}\left[\|\delta_i\|^2\right]\leq 4\rho^2\mathcal{R}^2 \text{ and } \mathbb{E}_{t_0}\left[\|\hat{\mathbf{x}}_{t_0+i}\|^2\right]\leq c\frac{\eta^2 r^2\beta_t^2}{d}\right)$$

$$\leq c\eta\rho\mathcal{R}\frac{\eta r}{\sqrt{d}}\mathbb{E}_{t_0}\left[\sum_{i=0}^{t-1}\frac{(1+\eta\gamma)^{t-1}}{\sqrt{\eta\gamma}}\right] \qquad \text{(Proposition B.14)}$$

$$\leq c\eta\rho\mathcal{R}\frac{\eta r}{\sqrt{d}}\mathcal{I}\beta_t \qquad \text{(Proposition B.14, another direction)}$$

$$\leq c\frac{\eta r\beta_t}{\sqrt{d}}(\eta\rho c_{\mathcal{R}}\sqrt{\frac{\varepsilon}{\rho}}\cdot\frac{c_{\mathcal{I}}}{\eta\sqrt{\rho\varepsilon}}) \qquad \left(\mathcal{R}=c_{\mathcal{R}}\sqrt{\frac{\varepsilon}{\rho}} \text{ and } \mathcal{I}=\frac{c_{\mathcal{I}}}{\eta\sqrt{\rho\varepsilon}}\right)$$

$$\leq cc_{\mathcal{R}}c_{\mathcal{I}}\frac{\eta r\beta_t}{\sqrt{d}},$$

and it suffices to choose $cc_{\mathcal{R}}c_{\mathcal{I}}\leq\frac{1}{40}$ so that $\mathbb{E}_{t_0}\left[\|\Delta_t\|\right]\leq\frac{1}{10}\mathbb{E}_{t_0}\left[\|\Xi_t\|\right]$.

**Bounding $\|\mathcal{E}_t+\eta\hat{\mathbf{e}}_{t_0+t}\|$.** **For a general compressor**, by Lemma B.16 we know that

$$\mathbb{E}_{t_0}\left[\|\mathcal{E}_t+\eta\hat{\mathbf{e}}_{t_0+t}\|^2\right]\leq\frac{cc_{\mathcal{I}}\eta^3(1-\mu)L^2\chi^2\beta_t^2}{\mu^2\sqrt{\rho\varepsilon}}$$

Using $\chi\leq 2r$, to show that $\mathbb{E}_{t_0}\left[\|\mathcal{E}_t+\eta\hat{\mathbf{e}}_{t_0+t}\|\right]^2\leq\mathbb{E}_{t_0}\left[\|\mathcal{E}_t+\eta\hat{\mathbf{e}}_{t_0+t}\|^2\right]\leq\frac{1}{100}\mathbb{E}_{t_0}\left[\|\Xi_t\|\right]^2$, it suffices to guarantee that

$$\frac{c_{\mathcal{I}}\eta^3(1-\mu)L^2\chi^2\beta_t^2}{\mu^2\sqrt{\rho\varepsilon}}\leq c\frac{\beta_t^2\eta^2 r^2}{d}\iff\eta\leq c\frac{\mu^2\sqrt{\rho\varepsilon}r^2}{c_{\mathcal{I}}d(1-\mu)\chi^2 L^2}$$

Using that $\chi^2\leq 2r^2$ for sufficiently large $c_r$, we have:

$$\eta\leq c\frac{\mu^2\sqrt{\rho\varepsilon}}{c_{\mathcal{I}}d(1-\mu)L^2}$$

**For a linear compressor**, By Lemma B.17 we have:

$$\frac{c_{\mathcal{I}}\eta^3(1-\mu)L^2\beta_t^2 r^2}{\mu^2 d\sqrt{\rho\varepsilon}}\leq c\frac{\beta_t^2\eta^2 r^2}{d}\iff\eta\leq c\frac{\mu^2\sqrt{\rho\varepsilon}}{c_{\mathcal{I}}(1-\mu)L^2}$$

**Bounding $\|Z_t\|$.** First, we consider the case when Assumption C doesn't hold (i.e. $\tilde{\ell} = +\infty$). Since $Z_t$ is the sum of independent random variables:

$$\mathbb{E}_{t_0}\left[\|Z_t\|^2\right] \leq \eta^2 \sum_{i=0}^{t-1}(1+\eta\gamma)^{2(t-i-1)}2\eta^2\sigma^2 \leq 2\eta^4\beta_t^2\sigma^2$$

To prove that $\mathbb{E}_{t_0}\left[\|Z_t\|\right]^2 \leq \mathbb{E}_{t_0}\left[\|Z_t\|^2\right] < \frac{1}{100}\mathbb{E}_{t_0}\left[\|\Xi_t\|\right]^2$, it suffices to show that

$$\eta^4\beta_t^2\sigma^2 \leq c\frac{\beta_t^2\eta^2r^2}{d} \iff \sigma\sqrt{d} \leq cr \iff \sigma^2 d \leq cc_r^2\frac{\varepsilon^2}{L\eta} \iff \eta \leq c\frac{c_r^2\varepsilon^2}{\sigma^2 L d},$$

which holds when $c_\eta \leq cc_r^2$, by Definition B.2.

Finally, we consider the case when Assumption C holds (i.e. $\tilde{\ell} < +\infty$). Since stochastic gradient is Lipschitz, we have $\|\hat{\zeta}_i\| \leq 2\tilde{\ell}\|\hat{x}_i\|$ and:

$$
\begin{aligned}
\mathbb{E}_{t_0}\left[\|Z_t\|^2\right] = \mathbb{E}_{t_0}\left[\|\eta\sum_{i=0}^{t-1}(I-\eta H)^{t-i-1}\hat{\zeta}_i\|^2\right] \quad &\text{(Definition B.11)}\\
\leq \eta^2 \sum_{i=0}^{t-1}\mathbb{E}_{t_0}\left[\|(I-\eta H)^{t-i-1}\hat{\zeta}_i\|^2\right] \quad &\text{(Noises are independent)}\\
\leq \eta^2 \sum_{i=0}^{t-1}\|(1+\eta\gamma)^{t-i-1}\|^2 \cdot \mathbb{E}_{t_0}\left[\|\hat{\zeta}_i\|^2\right] \quad &\text{(Since $\gamma$ is the smallest negative eigenvalue of $H$)}\\
\leq \eta^2 \sum_{i=0}^{t-1}\|(1+\eta\gamma)^{t-i-1}\|^2 \cdot \tilde{\ell}^2\mathbb{E}_{t_0}\left[\|\hat{\mathbf{x}}_{t_0+i}\|^2\right] \quad &\text{(Assumption C)}\\
\leq c\eta^2\mathcal{I}\frac{\eta^2r^2\beta_t^2}{d} \quad &\text{(See derivation for $\|\Delta_t\|$ above)}
\end{aligned}
$$

Therefore $\mathbb{E}_{t_0}\left[\|Z_t\|\right] \leq c\eta\tilde{\ell}\sqrt{\mathcal{I}}\frac{\beta_t\eta r}{\sqrt{d}}$. To guarantee that $\mathbb{E}_{t_0}\left[\|Z_t\|\right] \leq \frac{1}{10}\mathbb{E}_{t_0}\left[\|\Xi_t\|\right]$, it suffices to show that

$$\eta\tilde{\ell}\sqrt{\mathcal{I}}\frac{\beta_t\eta r}{\sqrt{d}} \leq c\frac{\eta r\beta_t}{\sqrt{d}} \iff \eta\tilde{\ell}\sqrt{\mathcal{I}} \leq c \iff \frac{c_{\mathcal{I}}^2\eta\tilde{\ell}^2}{\sqrt{\rho\varepsilon}} \leq c \iff \eta \leq c\frac{\sqrt{\rho\varepsilon}}{c_{\mathcal{I}}^2\tilde{\ell}^2},$$

which holds when $c_{\mathcal{I}}^2c_\eta \leq c$. $\qquad\qquad\square$

---

**Theorem B.19** *Under Assumptions A and B, for $\eta$ as in Definition B.2, after $T = \tilde{O}\left(\frac{f_{\max}}{\eta\varepsilon^2}\right)$ iterations of Algorithm 1 at least half of points $\mathbf{x}_0, \ldots, \mathbf{x}_T$ are $\varepsilon$-SOSP w.h.p. The condition in Line 5 is triggered at most $\tilde{O}(\frac{f_{\max}}{\mathcal{F}}) = \tilde{O}(\frac{T}{\mathcal{I}}) = \tilde{O}(\varepsilon^{-3/2})$ times.*

---

Note that the fraction of $\varepsilon$-SOSP can be made arbitrary close to 1: to achieve $1 - \delta$ fraction, we show that at most $\delta/2$ fraction has large gradients and $\delta/2$ fraction has Hessian with a large negative eigenvalue. For simplicity, in the theorem we consider $\delta = 1/2$.

**Proof :** As in the previous Lemma, $c$ is used to denote constants and may change its meaning from line to line. On the high level, the proof is the following: to show that at least half of points are $\varepsilon$-SOSP, it suffices to show that at most quarter of the points have large gradient, i.e. $\|\nabla f(\mathbf{x}_{t_0})\| \geq \varepsilon$, and we show that at most quarter of the points have escape directions, i.e. $\lambda_{\min}(\nabla^2 f(\mathbf{x}_{t_0})) < -\sqrt{\rho\varepsilon}$.

By Corollary A.8, by Markov inequality, when at least quarter of points have $\|\nabla f(\mathbf{x}_t)\| \geq \varepsilon$, then $f(\mathbf{x}_0) - f(\mathbf{x}_T) \geq f_{\max}$ with constant probability, which is impossible (setting the error to 0 can only decrease $\mathbb{E}\left[\|\mathbf{e}_t\|^2\right]$ and $\mathbb{E}\left[f(\mathbf{x}_t)\right]$, and therefore the Corollary holds).

It remains to show that there is at most quarter of points such that $\lambda_{\min}(\nabla^2(f(\mathbf{x}_t))) \leq -\sqrt{\rho\varepsilon}$. By Line 5 of Algorithm 1, and Proposition B.3, for any $t'$ there exists $t_0 \in [t' - \mathcal{I}, t']$ such that $\lambda_{\min}(\nabla^2 f(\mathbf{x}_{t_0})) < -\sqrt{\rho\varepsilon}/2$ and $\mathbf{e}_{t_0} = 0$. We show that when compressed SGD starts from $\mathbf{x}_{t_0}$, the objective improves.

**Proposition B.20** If $\lambda_{\min}(\nabla^2 f(\mathbf{x}_{t_0})) < -\frac{\sqrt{\rho\varepsilon}}{2}$, then for some $t \leq \mathcal{I}$:

$$f(\mathbf{y}_{t_0}) - \mathbb{E}_{t_0}\left[f(\mathbf{y}_{t_0+t})\right] \geq \mathcal{F}$$

**Proof :** By Lemma B.18, we bound $\hat{\mathbf{y}}_{t_0+t} = \mathbf{y}'_{t_0+t} - \mathbf{y}_{t_0+t}$ (Definition B.10):

$$\mathbb{E}_{t_0}\left[\|\hat{\mathbf{y}}_{t_0+t}\|^2\right] \geq c\frac{\beta_t^2\eta^2 r^2}{d} \geq c\frac{(1+\eta\gamma)^{2t}\eta^2}{d\eta\gamma} \cdot \frac{\varepsilon^2}{L\eta} \geq c\frac{(1+\eta\gamma)^{2t}\varepsilon^2}{d\gamma L}$$

For $t = \mathcal{I}$, we have $(1+\eta\gamma)^{\mathcal{I}} \geq (1 + \eta\sqrt{\rho\varepsilon})^{c\mathcal{I}/\eta\sqrt{\rho\varepsilon}} \geq e^{c\mathcal{I}}$. By selecting $c\mathcal{I} \geq c\log\frac{dL\rho2\mathcal{R}^2}{\sigma\varepsilon}$ for some $c$, we have $\mathbb{E}_{t_0}\left[\|\hat{\mathbf{y}}_{t_0+t}\|^2\right] \geq 2\mathcal{R}^2$, and therefore:

$$\mathbb{E}_{t_0}\left[\|\mathbf{y}_{t_0} - \mathbf{y}_{\mathcal{I}}\|^2\right] = \max(\mathbb{E}_{t_0}\left[\|\mathbf{y}_{t_0} - \mathbf{y}_{\mathcal{I}}\|^2\right], \mathbb{E}_{t_0}\left[\|\mathbf{y}_{t_0} - \mathbf{y}'_{\mathcal{I}}\|\right]^2) \geq \frac{1}{2}\mathbb{E}_{t_0}\left[\|\mathbf{y}'_{\mathcal{I}} - \mathbf{y}_{\mathcal{I}}\|^2\right] \geq \mathcal{R}^2$$

Since by Proposition B.9 $\mathbf{y}_{t_0+t}$ and $\mathbf{y}'_{t_0+t}$ have the same distribution, $\mathbb{E}_{t_0}\left[\|\mathbf{y}_{t_0} - \mathbf{y}_{\mathcal{I}}\|\right] = \mathbb{E}_{t_0}\left[\|\mathbf{y}_{t_0} - \mathbf{y}'_{\mathcal{I}}\|\right]$, and therefore

$$\mathbb{E}_{t_0}\left[\|\mathbf{y}_{t_0} - \mathbf{y}_{\mathcal{I}}\|^2\right] \geq \mathcal{R}^2,$$

and by Corollary B.6:

$$f(\mathbf{y}_{t_0}) - \mathbb{E}_{t_0}\left[f(\mathbf{y}_{t_0+\mathcal{I}})\right] \geq \mathcal{F}.$$

$\square$

*(Proof of Theorem B.19 continued)* Let $t_1, \ldots, t_k$ be the iterations where the condition at Line 5 of Algorithm 1 is triggered, and let $t_0 = 0$ and $t_{k+1} = T$. Consider iterations $i$ such that there exists $t' \in [t_i, t_{i+1})$ with $\nabla^2 f(\mathbf{x}_{t'}) < -\sqrt{\rho\varepsilon}/2$ and $\|\nabla f(\mathbf{x}_{t'})\| \leq \varepsilon$. Therefore, by smoothness, for any $t \in [t_i, t_{i+1})$ we have $\|\nabla f(\mathbf{x}_t)\| \leq \|\nabla f(\mathbf{x}_{t'})\| + \|\nabla f(\mathbf{x}_{t'}) - \nabla f(\mathbf{x}_t)\| \leq \varepsilon + 2L\mathcal{R}$. Moreover, since $\mathbf{x}_{t_{i+1}-1} \in B(\mathbf{x}_{t_i}, \mathcal{R})$, $\|\nabla f(\mathbf{x}_{t_{i+1}-1})\| \leq \varepsilon + 2L\mathcal{R}$ and by our choice of $\eta$, we have $\mathbf{x}_{t_{i+1}} \in B(\mathbf{x}_{t_i}, 2\mathcal{R})$.

By Line 6 of Algorithm 1, $f(t_{i+1}) \leq f(t_i)$ for any $i$. On the other hand, by smoothness:

$$f(\mathbf{x}_{t_{i+1}}) \geq f(\mathbf{x}_{t_i}) + \langle \nabla f(\mathbf{x}_{t_{i+1}}), \mathbf{x}_{t_{i+1}} - \mathbf{x}_{t_i}\rangle - \frac{L}{2}\|\mathbf{x}_{t_{i+1}} - \mathbf{x}_{t_i}\|^2$$

$$\geq f(\mathbf{x}_{t_i}) - 2(\varepsilon + 2L\mathcal{R})(3\mathcal{R}) - \frac{L}{2}9\mathcal{R}^2$$

$$\geq f(\mathbf{x}_{t_i}) - 10L\mathcal{R}^2$$

$$= f(\mathbf{x}_{t_i}) - \tilde{O}(\varepsilon)$$

Therefore, the objective decreases by at most $\tilde{O}(\varepsilon)$, and since $f(\mathbf{x}_{t_{i+1}}) < f(\mathbf{x}_{t_i})$, the variance of $\mathbb{E}_{t_i}\left[f(\mathbf{x}_{t_i}) - f(\mathbf{x}_{t_{i+1}})\right]$ is at most $\tilde{O}(\varepsilon)$.

Using the proposition above, we have:

$$\mathbb{E}\left[f(\mathbf{y}_0) - f(\mathbf{y}_T)\right] \geq \sum_i \mathbb{E}\left[f(\mathbf{y}_{t_i}) - \mathbb{E}_{t_0}\left[f(\mathbf{y}_{t_{i+1}})\right]\right],$$

with total variance $k\tilde{O}(\varepsilon^2)$. Therefore, by Chebyshev inequality, with constant probability we have

$$f(\mathbf{y}_0) - f(\mathbf{y}_T) \geq \frac{1}{4}\frac{T}{\mathcal{I}}\mathcal{F} \geq \frac{c}{4}\frac{f_{\max}\eta\sqrt{\rho\varepsilon}}{\eta\varepsilon^2}\sqrt{\frac{\varepsilon^3}{\rho}} \geq \frac{c}{4}f_{\max},$$

which is impossible by selecting a sufficiently large constant in the choice of $T$.

$\square$