# OpenReview forum: "Escaping Saddle Points with Compressed SGD"
_NeurIPS.cc/2021/Conference — NeurIPS 2021 Poster_

### Official Review · Reviewer_goPg · 2021-06-28

**Rating:** 6
**Confidence:** 2

**Summary:**

This paper presents the compressed stochastic gradient descent with random perturbations and the convergence analysis.  The results show that the proposed algorithm enjoys the same rate as the GD with random perturbations but can reduce communications.

**Limitations And Societal Impact:**

Yes.

**Main Review:**

Strengths:
1. The authors extended the gradient descent with random perturbations to the compressed version and applied it to the distributed setting. As far as I know, this is novel.

2.The algorithm scheme uses error feedback and random noise perturbation, which is more complicated than the previous scheme. The paper combines previous techniques and develops several novel ones.

3. The writing of this paper is very clear. The "sketch of the proof" helps a lot to get the difficulty in the analysis.

4. A very interesting is that the authors also present numerical results to demonstrate the efficiency after adding noise.  This is few in previous papers about random perturbations to escape saddle points.

Weakness: 1. Although the proof of the proposed algorithm is not obvious,   previous works (including [Chi Jin,Sai Praneeth Karimireddy]) have provided many methods. This paper should be more specific on the point, i.e., the novelty of the technique in the proof.  From my view, it seems not very difficult to get proof.  Thus, I hope the authors could answer this comment.

2.  In RANDOMK compressor, the total communication is dependent on the sparsity level. If $k=\epsilon^{\tau}$, the total communication is then$O(1/\epsilon^{4-\tau})$. Is this true？If it is true, why just set $\tau=3/4$?

I will raise the score if my concerns are addressed.

**Time Spent Reviewing:**

5

---

> ### Author Response · Authors · 2021-08-10
> **Official Response**
>
> Thank you for the comments! Regarding novelty, please refer to our separate comment “Novelty”. Your other comment is addressed below. Please, let us know if you have any further questions. In case our response does address your concerns we will appreciate it if you could raise your score as mentioned in your initial review.
>
>
> “In RANDOMK compressor, the total communication is dependent on the sparsity level. If $k=\epsilon^\tau$, the total communication is then $O(\frac{1}{\epsilon^{4−\tau}})$. Is this true？If it is true, why just set $\tau=3/4$?”
>
> Unfortunately, the total communication is not $\tilde O(\frac 1 {\epsilon^{4-\tau}})$. While decreasing the sparsity parameter $k$ decreases the per-iteration communication, it increases the number of iterations, which requires one to select the value of $k$ optimally. From our main result (Theorem 3.4 in the full version), by using $\mu=k/d$ and $1 - \mu \approx 1$, it follows that the number of iterations is
> $$\tilde O(\frac{\alpha}{\epsilon^4} + \frac{d}{k \epsilon^3} + \frac{d^2}{k^2 \epsilon^{2 + 1/2}})$$
>
> Since every iteration requires $O(k)$ communication, the total communication is:
> $$\tilde O(\frac{k \alpha}{\epsilon^4} + \frac{d}{\epsilon^3} + \frac{d^2}{k \epsilon^{2 + 1/2}})$$
>
> When $k$ is set to $\epsilon^\tau$, the total communication becomes:
> $$\tilde O(\frac{\alpha}{\epsilon^{4 - \tau}} + \frac{d}{\epsilon^3} + \frac{d^2}{\epsilon^{2 + 1/2 + \tau}})$$
>
> You can see that while the first term has the total communication $\tilde O(\frac 1 {\epsilon^{4-\tau}})$ (ignoring other terms except for $\epsilon$) as you specified, and therefore decreases with $\tau$, the third term increases with $\tau$. Therefore, it’s necessary to select $\tau$ which balances these terms. For $\tau = 3/4$, both terms are of order $\tilde O(\frac{1}{\epsilon^{3 + 1/4}})$.

---

> > ### Comment · Reviewer_goPg · 2021-08-18
> > **thanks**
> >
> > I thank the authors for their responses. After taking a look at the other reviews, I wish to keep my grade and confidence score.

---

### Official Review · Reviewer_VP57 · 2021-07-12

**Rating:** 6
**Confidence:** 5

**Summary:**

This paper is concerned with finding local minima, or second order stationary points of a function $\mathbb{R}^d \to \mathbb{R}$. The key contribution of the paper is to combine the ideas of stochastic gradient descent (SGD) and compressed gradient descent (CGD). This leads to an algorithm which takes care of both the convergence rate and communication complexity, and is favorable for distributed computing. Theoretical results of the convergence and communication complexity are rigorously established, and some empirical studies are conducted.

**Limitations And Societal Impact:**

The authors have adequately addressed the limitations and potential negative societal impact of their work.

**Main Review:**

The idea of combining SGD and CGD might not be too novel, since there are works using these two ingredients to find the first order stationary points. However, this paper is the first instance to study rigorously finding the second order stationary points, a.k.a. escaping the saddle points, using the compressed SGD. This is important for learning problems, since the main obstacle is saddle points not the local extrema. The paper is well-written, and the results seem to be scientifically correct. However, there are still a few unclear points to me (see below), and I am not sure what is exactly the technical contribution of this paper (i.e. compare to previous works of compressed SGD applied to the first order optimization). I would consider to raise my score if the following points are addressed.

Unclear points and suggestions:
1. l.72 and elsewhere: there is the notation $\tilde{\mathcal{O}}$ which is not explained. I understand that in other previous papers this notation is also used to hide log term, but it might be better to explain for ease of reading.

2. l.79-83: It is unclear to me what it means "RANDOMK preserves k random coordinates", does it mean $\mathcal{C}(x)$ is $k$-dimensional? This way, how to make sense of $x - \mathcal{C}(x)$? ($x$ is $d$-dimension, while $\mathcal{C}(x)$ is $k$-dimension). The authors should explain this in a clear way.

3. Theorem 1.1: the rate of convergence is $\tilde{\mathcal{O}}(\varepsilon^{-4})$ which is slower than $\tilde{\mathcal{O}}(\varepsilon^{-2})$ established in [24, 25]. The authors may comment on this fact. Any explanation?

4. l.130-133: do you mean $\theta_i \sim \mathcal{D}_i$ in the definition of $f_i(x)$? Otherwise, the authors should make clear the definitions of $f_i, F_i, \theta_i$ and connections.

5. l.188: I guess the authors mean Line 6 of Algorithm 1 instead of Line 1 of Algorithm 6.

6. Theorem 3.1: I am a bit confused with the notations. It seems to me that $\mathcal{R}$ is a positive number as the distance in Euclidian space. But in the theorem, $t \in \mathcal{R}$ is used as an index set for the time $t$.

7. Theorem 3.1 and 3.2: in both theorems, there is a claim that at least half of visited points are $\varepsilon$-SOSP. I wonder if this set is finite or infinite. My understanding is that this set would be infinite, so the authors may be a bit careful about what "at least half" mean (there is possibly a limiting frequency involved).

8. l.239: (JACM) should be removed.

9. What is the technical contribution of this paper in comparison with the previous ones on the first order optimization?

10. There is also some work on the perturbed gradient descent, with non-uniform perturbation, see e.g. Perturbed gradient descent with occupation time (https://arxiv.org/abs/2005.04507v1). Theoretically the proposed algorithm achieves at least the same convergence rate as in [24, 25]. They also showed in practice, the carefully chosen perturbation is efficient in escaping saddle points even with some bad initializations.

**Time Spent Reviewing:**

1 hour

---

> ### Author Response · Authors · 2021-08-10
> **Official Response**
>
> Thanks a lot for your detailed comments. We’ve addressed your questions through a number of edits as described below. Please, let us know if you have any further questions. In case our edits do address your concerns we will appreciate it if you could raise your score as mentioned in your initial review.
>
> 1) Great point, thanks for pointing this out. The definition of $\tilde O$ has been accidentally dropped in the editing process. This notation hides polynomial dependence on $L$, $\rho$, $f_{max}$, $\sigma$, $\tilde \ell$ and polylogarithmic dependence on all parameters (as is standard in the literature, see e.g. [19,25]). We put the discussion of this notation back in place before Theorem 1.1.
> 2) For Random-k, all coordinates except for the $k$ randomly chosen ones are $0$. I.e. the result is a $d$-dimensional vector, with $k$ coordinates being the same as in $x$ and $d-k$ coordinates being $0$. Hence $x−C(x)$ is also a d-dimensional vector. We clarified this in the paper.
> 3) $O(\epsilon^{-2})$ convergence rate is achievable only when an exact gradient oracle is available. With only stochastic gradients available, $O(\epsilon^{-4})$ is a standard convergence rate. E.g. in [25], this is the difference between Theorems 13 (exact) and 16 (stochastic). In our paper, we consider the stochastic gradient case and hence our results should be compared with Theorem 16 in [25] which also has $O(\epsilon^{-4})$ convergence rate.
> 4) Thank you, it should be $\theta_i$. Fixed.
> 5) Thank you, fixed.
> 6) Thanks, this is an unfortunate notation clash. In Algorithm 1, $\mathcal R$ indeed is a positive number. In Theorem 3.1, $\mathcal R$ is the set of all iterations where the condition from Line 5 is triggered. In Theorem 3.1 we changed the notation from $\mathcal R$ to $\mathcal S$.
> 7) Thanks, we believe the confusion might be due to our unfortunate wording, which is easy to misinterpret. We’ve edited the statements to avoid confusion. In both Theorem 3.1 and Theorem 3.2 the set of visited points is finite since we only consider the first T iterations (as in the description of Algorithm 1).
>     * In Theorem 3.2, after these $T$ iterations of compressed SGD are complete, among points $x_1, \ldots, x_T$, at least $T/2$ of them are $\epsilon$-SOSP.
>     * In Theorem 3.1 the statement is more subtle. Let $\mathcal R$ be the set of iterations prior to the iteration $T$ such that the condition in Line 5 is triggered. At least half of the points visited in these iterations is guaranteed to be an $\epsilon$-SOSP.
> 8) Fixed.
> 9) Please refer to the separate comment “Novelty”.
> 10) Thanks, this is a very interesting reference, we added it to our citations.

---

> > ### Comment · Reviewer_VP57 · 2021-08-31
> > **Comment**
> >
> > I would like to thank the authors for the detailed response. I am pretty convinced, and I still stick to the original score.

---

### Official Review · Reviewer_XCGn · 2021-07-20

**Rating:** 7
**Confidence:** 3

**Summary:**

It has been proven in previous works that SGD can escape saddle points existing in non-convex objective functions and find local minima. This paper considers non-convex optimization problems in the distributed setting where the gradients communicating between workers and the parameter server are compressed to reduce the communication cost. It shows that when the compression technique is either \mu-compression or Random-K compression then SGD can still escape from saddle points. Their analysis shows that under some reasonable assumptions the compressed SGD can reach \epsilon-SOSP after O(\epsilon^{-4}) iterations.

**Limitations And Societal Impact:**

Yes

**Main Review:**

This work is an incremental work that is based on the analysis of compressed SGD and saddle point avoidance properties of SGD that have been shown in previous works. However, since in practice training big models using distributed settings is essential, this work provides some guarantee about finding second-order stationary points while SGD is used.

It is easy to follow the main text and also the proofs. They considered various scenarios and analyzed them carefully to improve the quality of their work.



My questions

- I was wondering how hard it is to generalize this analysis for applying a general random function to a stochastic gradient. For example, should this function be an unbiased estimator?

- In Fig 2 it is not clear why you claim that SGD encountered and then escaped from a saddle point. Since the x-axis is based on the epoch, SGD has the ability to escape from a saddle point even after a few iterations. Also for the graph in the 2nd row, it would be more informative to depict the gradient norm at the current iteration, not the averaged one.


**Time Spent Reviewing:**

4

---

> ### Author Response · Authors · 2021-08-10
> **Official Response**
>
>
> Thank you for the comments! We hope that our response below addresses your concerns. Please, let us know if you have any further questions!
>
> Regarding applying a general random function: can we ask you please to elaborate on the question? If the random function is unbiased and has bounded variance, then one can apply the standard SGD analysis. If it’s not unbiased and it’s not a compressor per Definition 2.1, then it’s hard to provide any guarantees: e.g. a function can return a constant zero vector (meaning $\mu=0$ in Definition 2.1).
>
> Figure 2 shows that at some point the gradient norm becomes 0, while the function value is far from optimal, and we verified that the Hessian has a negative eigenvalue with a very small absolute value. As you correctly point out, in practice, SGD can escape from saddle points (and we can see this in Figures 2a and 2d). However, the number of iterations required for escaping depends on the properties of both the saddle point (namely, the smallest eigenvalue of the Hessian at this point) and the stochastic oracle (in particular, its variance). While usually SGD immediately escapes saddle points, in this scenario it happens slowly, in particular, because the smallest eigenvalue of the Hessian is close to 0. Our experiments show that adding noise facilitates the escaping behavior significantly.
>
> Regarding averaged gradients in Figure 2: thank you for pointing it out! Since it’s computationally infeasible to compute exact gradients at every point, we use stochastic gradients, which have large fluctuations. To reduce this variance, we average the gradients over the last 100 iterations. For a sufficiently small step size, the iterates are close to each other, resulting in a good approximation. We’ve added a clarification that we average stochastic gradients, not the exact gradients. In the appendix, we’ll add the exact gradient norms computed at the end of every epoch.

---

### Author Response · Authors · 2021-08-10
**Novelty**

The technical contributions of the paper are: 1) the algorithm and 2) analysis showing that the algorithm converges to a saddle point (see Section 3.3 of the full version).

1) Algorithm. Our algorithm periodically resets the error to 0 and, if the function doesn’t significantly improve after a fixed number of iterations, discards these iterations. To the best of our knowledge, the idea of error reset is novel.

2) Analysis. For the general compressor, we consider large and small-gradient cases separately. This is a significant deviation from the previous analysis  [25], and is crucial for deriving the bounds. Furthermore, the bound on the deviation from a quadratic function arising from the compression error (Lemmas 16 and 18) is entirely new.

---

### Decision · Program_Chairs · 2021-09-27

**Decision:**

Accept (Poster)

**Comment:**

The paper generated a fair bit of discussion, and the reviewers agree that the paper has interesting approach and combination of arguments of escaping saddle point with compressed gradient is very elegant.